# Controlled human hookworm infection remodels plasmacytoid dendritic cells and regulatory T cells towards profiles seen in natural infections in endemic areas

Mikhael D. Manurung [1,5], Friederike Sonnet[1,5], Marie-Astrid Hoogerwerf [1], Jacqueline J. Janse[1], Yvonne Kruize[1], Laura de Bes-Roeleveld[1], Marion König [1], Alex Loukas[2], Benjamin G. Dewals [3], Taniawati Supali[4], Simon P. Jochems [1], Meta Roestenberg [1,6], Mariateresa Coppola [1,6] & Maria Yazdanbakhsh [1,6] ✉

Hookworm infection remains a significant public health concern, particularly in low- and middle-income countries, where mass drug administration has not stopped reinfection. Developing a vaccine is crucial to complement current control measures, which necessitates a thorough understanding of host immune responses. By leveraging controlled human infection models and high-dimensional immunophenotyping, here we investigated the immune remodeling following infection with 50 *Necator americanus* L3 hookworm larvae in four naïve volunteers over two years of follow-up and compared the profiles with naturally infected populations in endemic areas. Increased plasmacytoid dendritic cell frequency and diminished responsiveness to Toll-like receptor 7/8 ligand were observed in both controlled and natural infection settings. Despite the increased CD45RA+ regulatory T cell ($T_{regs}$) frequencies in both settings, markers of $T_{regs}$ function, including inducible T-cell costimulatory (ICOS), tumor necrosis factor receptor 2 (TNFR2), and latency-associated peptide (LAP), as well as in vitro $T_{regs}$ suppressive capacity were higher in natural infections. Taken together, this study provides unique insights into the immunological trajectories following a first-in-life hookworm infection compared to natural infections.

Human hookworm infection continues to cause significant health and economic burdens in low- and middle-income countries[1]. To reduce the morbidity caused by hookworm infection, mass drug administration (MDA) with albendazole or mebendazole has been implemented[2]. However, MDA does not provide the necessary protection against reinfection to break the transmission cycle. Furthermore, concerns about treatment failure with mebendazole and the possible emergence of drug resistance have been raised[3], highlighting the need for a vaccine.

The rational design of vaccines requires a thorough understanding of the immune response of the host. Controlled human

[1]Leiden University Center for Infectious Diseases (LU-CID), Leiden University Medical Center, Leiden, The Netherlands. [2]Centre for Molecular Therapeutics, Australian Institute of Tropical Health and Medicine, James Cook University, Cairns, Australia. [3]Laboratory of Immunology-Vaccinology, FARAH, University of Liège, Liège, Belgium. [4]Department of Parasitology, Faculty of Medicine, University of Indonesia, Jakarta, Indonesia. [5]These authors contributed equally: Mikhael D. Manurung, Friederike Sonnet. [6]These authors jointly supervised this work: Meta Roestenberg, Mariateresa Coppola, Maria Yazdanbakhsh. ✉e-mail: m.yazdanbakhsh@lumc.nl

infection models serve as unique and promising means to study the host's immune response, owing to the known time of infection and inoculum dose, which are difficult to determine in naturally occurring infections in endemic areas. The combination of controlled human infection models with high-dimensional immunophenotyping has revealed new insights into host-pathogen interactions, such as in malaria[4,5]. Although controlled human hookworm infection (CHHI) studies have involved more than 200 participants in total[6], a high-dimensional analysis of the immune response to hookworm has not been performed.

In CHHI studies conducted to date, eosinophilia, the production of total and hookworm-specific immunoglobulin (Ig) E and IgG, as well as cytokine production by immune cells in culture supernatants have been reported, indicating the activation of CD4[+]T cells and mixed T helper 1 ($T_H$1) or 2 ($T_H$2) cytokine responses[7,8]. Here, we performed a high-dimensional mapping of cellular immunological responses in CHHI study participants who were experimentally infected with one dose of 50 *Necator americanus* L3 larvae and followed for two years[9]. Changes in the profile of peripheral blood mononuclear cells (PBMC) and intracellular cytokine production were analyzed. In addition, we examined the immune profiles of hookworm-infected Indonesians living in Flores, to see if our findings in controlled infection can be generalized to naturally occurring infections in an endemic setting.

Our study showed that controlled hookworm infection with a single dose of larvae caused numerous changes in the circulating immune cells, such as plasmacytoid dendric cells and regulatory T cells ($T_{regs}$), that showed linked trajectories and mimicked, to some degree, what is seen in naturally occurring infections. However, clear differences were also seen, which could be explained by the repeated exposure to hookworms and coinfections that might exert extra modulatory effects on the immune system.

## Results

### Patent infection in all participants after controlled hookworm infection

To investigate immunological changes after primary hookworm infection, four healthy hookworm-naïve volunteers were enrolled in the Controlled Human Hookworm Infection in Leiden (CHHIL) study. As previously reported[9], the CHHIL study participants were infected with one dose of 50 infectious *N. americanus* L3 larvae and followed for two years. Peripheral blood mononuclear cells (PBMC) were isolated at seven time points (Fig. 1A). All four volunteers started excreting eggs between 8 and 10 weeks post-infection (WPI) as measured by the Kato-Katz technique (Fig. 1B). Eosinophil counts increased significantly starting at 3 WPI ($P = 0.042$, compared to baseline), peaked at 6 WPI (range, $2.02 \times 10^9 - 6.96 \times 10^9$ eosinophils/L, $P = 0.0001$), and subsequently decreased from 12 WPI but remained significantly elevated for two years (Fig. 1C, $P < 0.018$)[9]. Adverse events were generally mild, with skin itching and rash at the site of larvae inoculation occurring shortly after the infection, and abdominal adverse events (flatulence, nausea, and abdominal cramping) were reported between 3 and 9 WPI[9]. These results indicate that the controlled hookworm infection resulted in a patent and tolerable infection in all participants during the two-year study period.

### Hookworm infection-driven changes in immunological profiles

To characterize changes in immune cell composition after primary hookworm infection, PBMC from CHHIL participants were analyzed using mass cytometry with a 37-marker antibody panel (Table S1). Using FlowSOM clustering on the acquired cells, we obtained 45 immune cell clusters from a total of 7.8 million immune cells. The FlowSOM clusters were classified into eight immune cell lineages based on marker expression: conventional CD4[+] and CD8[+] αβ T cells,

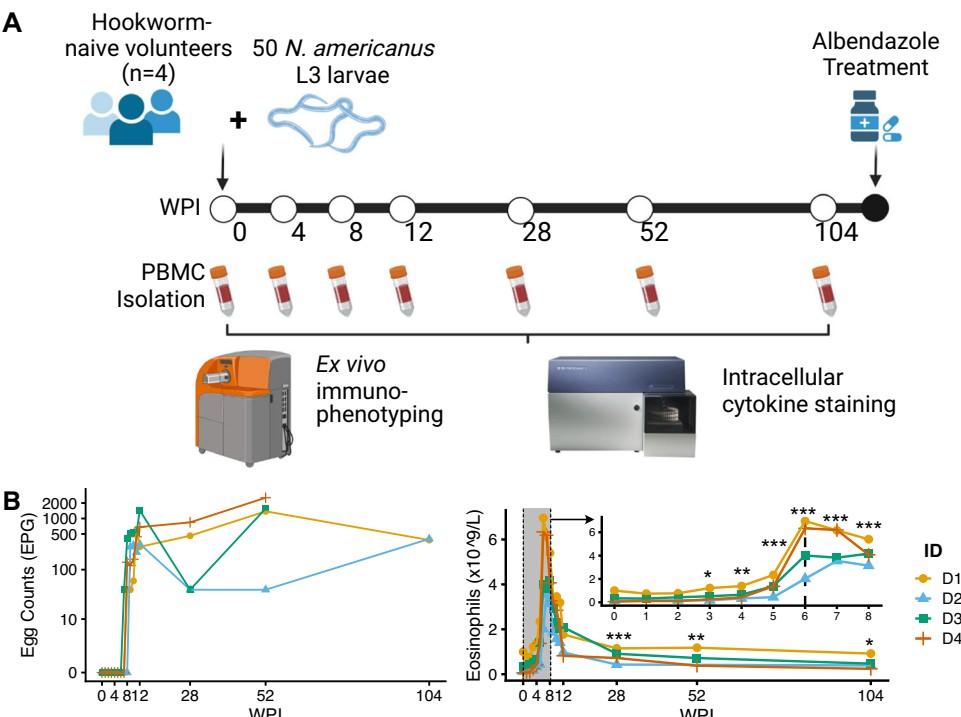

**Fig. 1 | The Controlled Human Hookworm Infection in Leiden (CHHIL) study. A** Study timeline for the hookworm challenge, PBMC isolation, termination of infection with albendazole, and cytometry data acquisition. This figure panel is created with BioRender.com under a Creative Commons Attribution-NonCommercial-NoDerivs 4.0 International license. **B** Left: Eggs per gram of stool (EPG) as measured by Kato-Katz. Right: Eosinophil counts (x 10⁹/L) from 0 to 104 WPI. The inset plot shows weekly measurements of eosinophils within the first 8 WPI. Changes in eosinophil counts relative to baseline (0 WPI) were tested using a gaussian linear mixed model with Dunnett's test. EPG, egg per gram; WPI, week(s) post-infection. *$P < 0.05$, **$P < 0.01$, ***$P < 0.001$. Source data are provided as a Source Data file.

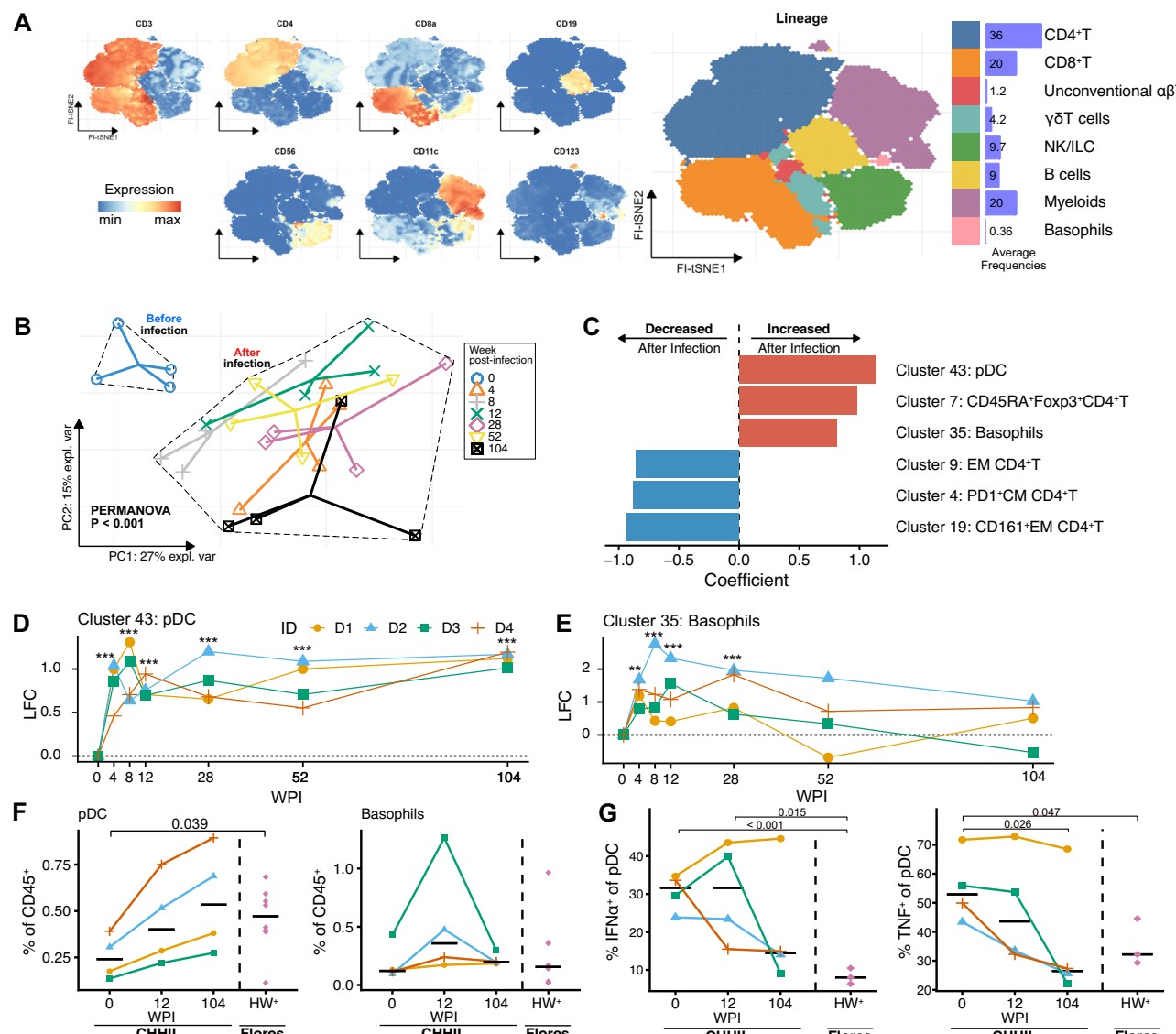

**Fig. 2 | Mass cytometry reveals global changes in the signature of immune cells after a single infection with 50 *Na*-L3 larvae. A** Fl-tSNE projection of all live, singlet, CD45+ cells, colored by lineage marker expression levels (left panel) or colored by immune cell lineage annotations (right panel). The horizontal barplots indicate the average frequencies of the immune cell lineages relative to total CD45+ cells across samples. **B** Principal component analysis of immune cell clusters frequencies before (0 WPI) and after infection (4, 8, 12, 28, 52 and 104 WPI). Cell cluster counts were centered log-ratio (CLR) transformed, then within-subject mean centered to take repeated measures into account, and finally scaled to unit variance. The first two principal components explained 42% of the variance in the data. PERMANOVA P-value comparing sample centroids before and after infection is shown. Distinct shapes and colors indicate samples from the four donors for each time point. Solid lines connect the observations toward the centroid for each time point. **C** PERMANOVA coefficient of multivariate comparison of immune cell cluster composition before and after infection as shown in (**B**). The top three clusters that contribute the most to the differences for each direction of change are shown. **D**, **E** Log₂ fold change (LFC) in cluster frequencies (% of CD45+) for clusters 43 and

35 that are annotated as (**D**) plasmacytoid dendritic cells (pDC), and (**E**) basophils, respectively. Colors and shapes indicate a distinct individual (ID). Lines connect paired samples. ** FDR < 0.01, *** FDR < 0.001. **F** Frequencies of manually gated pDC (left) and basophils (right) in CHHIL and helminth-infected Flores donors. Horizontal lines indicate the median for each timepoint or group. Dashed vertical lines demarcate CHHIL (right) and hookworm-infected Flores individuals (HW+ Flores) (left). The changes in cell abundance were tested using a binomial generalized linear mixed effects model. **G** Frequencies of IFNα- and TNF-producing pDC after in-vitro stimulation with R-848, a TLR-7/8 agonist in CHHIL and Flores subjects. The cytokine frequencies were background subtracted. Differences in cytokine-producing cell frequencies were tested using a linear mixed effects model. *Na*-L3 Necator americanus L3-larvae, Fl-tSNE FFT-accelerated Interpolation-based t-SNE, NK natural killer cells, ILC innate lymphoid cells, PC Principal component, pDC plasmacytoid dendritic cells, EM effector memory, CM central memory, LFC Log₂ fold-change, WPI week(s) post-infection, CHHIL Controlled Human Hookworm Infection in Leiden, R-848 resiquimod. Source data are provided as a Source Data file.

unconventional αβ T cells, γδ T cells, natural killer and innate lymphoid cells (NK/ILC), B cells, myeloid cells, as well as basophils (Fig. 2A, Table S2); two clusters (clusters 25 and 41) were excluded from further analysis due to lack of signal for any lineage markers. Altogether, there were few significant changes in the frequency of cells at the lineage level (Fig. S1, Supplementary Data 1). However, when analyzed on cluster level, principal component analysis (PCA) revealed highly

significant segregation of the immune profiles before and after hookworm infection (Fig. 2B, PERMANOVA *P* < 0.001). This segregation was largely driven by increased frequencies of plasmacytoid dendritic cells (pDC, cluster 43), CD45RA+Foxp3+CD4+T cells (resting T regs, cluster 7), and basophils (cluster 35) as well as decreased frequencies of memory CD4+T cells (clusters 4, 9, and 19) after infection (Fig. 2C). Differential abundance analysis of the cell clusters revealed that changes in

FlowSOM cluster frequencies can be found as early as 4 WPI and persist up to 104 WPI (Fig. S2, Supplementary Data 2). These findings show that primary hookworm infection caused changes, starting early after infection, in the frequencies of immune cell clusters driven, among others, by the enrichment of CD45RA$^+$ resting T$_{regs}$, pDC, and basophils.

## Long-term expansion of pDC and basophils following hookworm infection

Among CHHIL participants, we found that pDC frequencies (cluster 43) increased significantly from 4 WPI, and remained elevated relative to baseline until year 2 (Fig. 2D), while basophil frequencies (cluster 35) increased from 4 WPI to 28 WPI and then normalized (Fig. 2E). We then asked to what extent changes in pDC and basophil frequencies after primary hookworm infection lasting two years, mirror what is seen in individuals with chronic hookworm infection residing in endemic areas. To this end, we compared our findings in the CHHIL participants with those seen in Indonesians living in Flores, an area endemic for soil-transmitted helminths, that had hookworm infection with or without other soil-transmitted helminths (HW$^+$; Table S4)[10]. To allow comparisons across datasets, we manually gated pDC and basophils from the CHHIL participants and HW$^+$ Flores residents (Fig. S3). The frequencies of pDC and basophils in HW$^+$ Flores residents were comparable to those seen in the CHHIL participants two years post-infection (Fig. 2F). We also tested the functional responses of pDC by stimulation with R-848 (Resiquimod), a TLR-7/8 ligand, and measurement of intracellular cytokine production (Fig. S4A, Table S3). The analysis revealed that the frequencies of IFNα- and TNF-producing pDC decreased over time in the CHHIL participants and, at two years post-infection, frequencies approached those seen in HW$^+$ Flores residents (Fig. 2G), except for one outlying CHHIL donor (D1). These findings indicate that chronic, two-year exposure to a primary hookworm infection leads to similar changes in pDC frequency and functionality as seen in hookworm-infected subjects in endemic areas.

## Hookworm infection induces the rearrangement of the T$_{regs}$ compartment

Modulation of host T$_{regs}$, which can promote the survival of helminths within their host[11,12], is one of the hallmarks of helminth infections[10]. Therefore, the fact that changes in immune profiles after controlled human hookworm infection were driven, among others, by T$_{regs}$, provided the opportunity to examine these cells together with data from individuals living in helminths-endemic areas, to see how closely changes in T$_{regs}$ after primary hookworm infection resembled those seen in endemic areas. To this end, T$_{regs}$ from CHHIL participants and HW$^+$ Flores residents before and after one year of deworming (Fig. 3 A) were manually gated from mass cytometry datasets (Fig. S3), integrated with cyCombine, and used as input for FI-tSNE dimensionality reduction as well as clustering using Cytosplore, which enables interactive exploration of mass cytometry datasets and identification of rare cell clusters (Fig. 3B).

We then examined changes in T$_{regs}$ composition after primary hookworm infection. Total T$_{regs}$ frequencies did not change significantly two years after infection, despite a significant decrease at 52 WPI (Fig. S5 A, $P = 0.024$). However, further analysis of T$_{regs}$ cluster composition using multilevel PCA revealed segregation of the T$_{regs}$ profile before and after infection (Fig. 3C, PERMANOVA $P < 0.001$). The segregation of the T$_{regs}$ profile was driven by the increase of resting T$_{regs}$ clusters (T$_{regs}$1 and 2) and effector T$_{regs}$ cluster 4 (CTLA4$^+$ICOS$^+$CD38$^{hi}$) frequencies, as well as the decrease of the remaining effector T$_{regs}$ clusters (T$_{regs}$5, 9, 10) (Fig. 3D and Fig. S5B). The changes in the resting and effector T$_{regs}$ clusters could occur as early as 4 WPI and persist up to 104 WPI (Fig. 3D and Fig. S5C; Supplementary Data 3).

Within the time points measured after hookworm infection, we found that the T$_{regs}$ profile at 4 WPI segregated from the other time points (Fig. S5D). At this time point, the frequencies of ICOS-expressing effector T$_{regs}$ cluster 4 were increased (Fig. 3D). Knowing that pDC can induce ICOS expression on T$_{regs}$[13], we found a strong positive correlation between pDC frequencies and T$_{regs}$ expressing ICOS (Fig. 3E, Pearson r = 0.84, $P = 0.009$). Given the distinctiveness of the T$_{regs}$ profile at 4 WPI, we also considered the possibility that the clustering-based analysis might obscure fine-grained changes occurring within this time frame. To this end, we used Milo to perform differential abundance analysis on partially overlapping neighborhoods of cells[14], enabling such analysis without having to discretize cells into computationally driven clusters. Performing Milo analysis on the T$_{regs}$ data was feasible, unlike for pDC and basophils, as the number of markers in our mass cytometry panel unraveled sufficient heterogeneity for T$_{regs}$. This analysis showed an enrichment of neighborhoods of T$_H$2-like T$_{regs}$ cluster 7 (Fig. 3F).

As noted in the changes post-infection (Fig. S2 and Supplementary Data 2), the frequencies of T$_H$2 cells decreased between 0 and 4 WPI (Table S2, cluster 16, FDR = 0.093), in parallel with the enrichment of T$_H$2-like T$_{regs}$. As activated non-regulatory CD4$^+$T cells can transiently express Foxp3[15,16], we asked whether T$_H$2 and T$_H$2-like T$_{regs}$ form a continuum of closely related subpopulations, or instead represent two well-separated clusters. To this end, the mass cytometry data of T$_H$2 and T$_H$2-like T$_{regs}$ were fed into PHATE dimensionality reduction, which preserves patterns in the data, such as trajectories and clusters[17]. We found that the T$_H$2 and T$_H$2-like T$_{regs}$ formed a continuum of cells and that the trajectory structure of the latter was associated with CD38 expression, a marker of T cell activation (Fig. 3G). These results suggest that the T$_H$2-like T$_{regs}$ are phenotypically related to T$_H$2 cells and raise the possibility that the former might arise from the latter upon activation.

Taken together, in-depth analysis captured dynamic processes taking place within four weeks of infection. pDC may interact with T$_{regs}$ leading to upregulation of ICOS expression, proposed to be associated with enhanced T$_{regs}$ function, while enriched neighborhoods of T$_H$2-like T$_{regs}$ could be found that seem to arise from related activated T$_H$2 cells.

To answer the question of whether T$_{regs}$ phenotype changes seen upon controlled hookworm infection resemble what is seen in hookworm-infected subjects from endemic areas, we compared changes in T$_{regs}$ found in CHHIL donors with those seen in HW$^+$ Flores residents before and after one year of deworming. The increase in frequencies of resting T$_{regs}$ in CHHIL donors (Fig. 3H, CHHIL) mirrored their decrease upon deworming in Flores residents (Fig. 3H, Flores). Thus, higher resting T$_{regs}$ frequencies were associated with hookworm infection in both controlled and endemic settings. There were, however, also differences in T$_{regs}$ phenotype and function between the two cohorts. The frequency of effector T$_{regs}$ cluster 4 (CCR7$^+$CTLA4$^+$ICOS$^+$CD38$^{hi}$) was higher in HW$^+$ Flores residents compared to the average of post-infection CHHIL time points (4 to 104 WPI) (Fig. 3I, $P = 0.029$) and decreased after deworming ($P = 0.021$). To determine differences in T$_{regs}$ functional capacity, we analyzed the surface expression of tumor necrosis factor receptor 2 (TNFR2) and TGF-β1 latency-associated peptide (LAP) on T$_{regs}$ and also the percentage of suppression of responder T cell (T$_{resp}$) proliferation with T$_{regs}$ suppression assay. The frequencies of T$_{regs}$ expressing TNFR2 and LAP were higher in HW$^+$ Flores residents than in CHHIL participants at 104 WPI (Fig. 3J and Fig. S4B). The percentage suppression by T$_{regs}$ was higher in HW$^+$ Flores residents compared to CHHIL donors, particularly at a 1:1 T$_{resp}$:T$_{regs}$ ratio (Fig. 3K, $P = 0.028$; Fig. S5E).

Taken together, we found that primary hookworm infection lasting two years caused a significant remodeling of the T$_{regs}$ compartment, as evidenced by increased frequencies of resting T$_{regs}$, which was paralleled in the profile seen in endemic areas, yet markers of T$_{regs}$

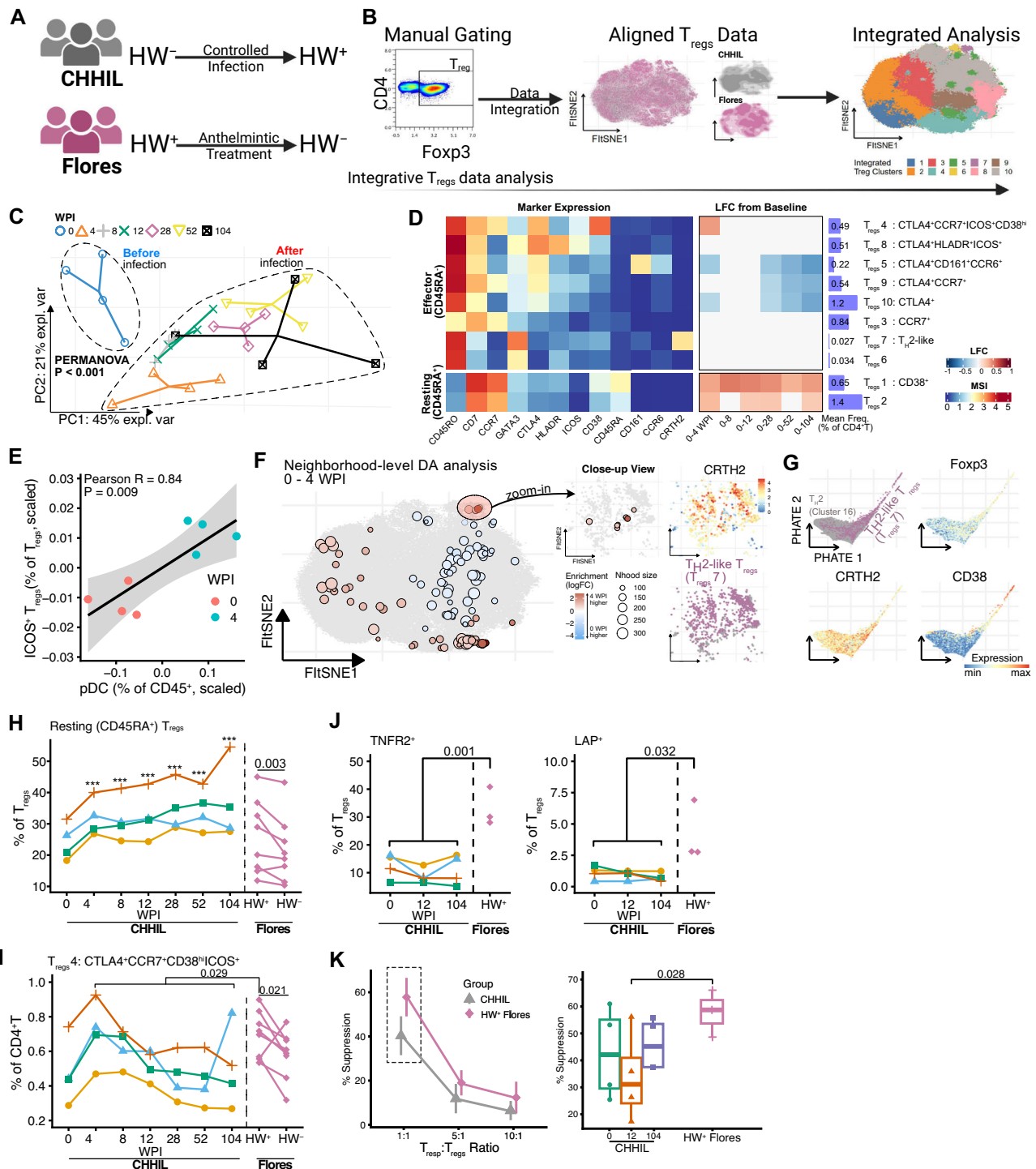

function were distinct, suggesting that $T_{regs}$ from endemic settings have stronger suppressive activity.

**Development of hookworm-specific CD4⁺ T cells responses**

Within the CD4⁺ T cell compartment of CHHIL participants, ten clusters changed significantly in frequencies throughout the infection (Fig. 4A). These clusters included naïve and memory CD4⁺ T clusters, as well as $T_H1$ and $T_H2$ cell subsets. Regarding the naïve and memory CD4⁺ T cells, we observed that a naïve cluster (cluster 3) increased in frequency whereas memory CD4⁺ T clusters (clusters 9, 13, 14, 17, 19) decreased within 4 WPI (Fig. 4A), which might reflect a response to an inflammatory process. We then mapped the development of antigen-specific responses by analyzing the frequencies of CD4⁺ T cells

producing $T_H1$ or $T_H2$ cytokines upon stimulation with hookworm crude antigen extract (Fig. S4C). At baseline and 4 WPI, the CD4⁺ T cells cytokine responses to crude larval antigen of the rodent hookworm *Nippostrongylus brasiliensis* were negligible (Fig. 4B). At 8 WPI, cytokine-producing cells were detected and the responses peaked at week 12 (Fig. 4B). After 104 WPI, the frequency of IL-4/IL-5/IL-13-producing CD4⁺ T cells remained detectable for three out of three donors, while IFNγ and TNF responses contracted after 28 WPI in all CHHIL participants.

To investigate changes in immune signature preceding the development of the antigen-specific response, we examined the phenotype of CD4⁺ T cells expressing CD38, which can indicate recently activated memory T cells[18]. To this end, we manually gated

**Fig. 3 | Helminths infection is associated with CD45RA⁺ resting T$_{regs}$ in both controlled and natural infection settings. A** Schematic of hookworm infection status (HW⁺) with respect to controlled infection or anthelmintic treatment in CHHIL participants or Flores residents, respectively. This figure panel is created with BioRender.com under a Creative Commons Attribution-NonCommercial-NoDerivs 4.0 International license. **B** Schematic of integrative T$_{regs}$ data analysis pipeline. Left: T$_{regs}$ were manually gated from mass cytometry data of CHHIL participants and Flores residents. Center: FI-tSNE projection of mass cytometry data alignment using cyCombine. Colors indicate the study of origin. Right: overlay of T$_{regs}$ clusters, which was obtained using Cytosplore, on integrated data embedding. **C** Principal component analysis of T$_{regs}$ cluster frequencies before (0 WPI) and after infection (4, 8, 12, 28, 52 and 104 WPI). Cell cluster frequencies were calculated relative to total CD4⁺T cells, within-subject mean-centered to take into account repeated measures, and scaled to unit variance. The first two principal components explained > 65% of the variance in the data. Distinct shapes and colors indicate samples from the four donors for each time point. PERMANOVA P-value comparing T$_{regs}$ profile before and after infection is shown. **D** Heatmaps summarising the changes in T$_{regs}$ cluster frequencies after infection with corresponding marker expression. Left: Heatmap panel showing the median signal intensity of T$_{regs}$ clusters marker expressions. The T$_{regs}$ clusters were classified as either resting T$_{regs}$ (CD45RA⁺) or effector T$_{regs}$ (CD45RA⁻). Center: Log fold changes in T$_{regs}$ cluster frequencies (as % of total CD4⁺T cells) relative to baseline (0 WPI). Changes in T$_{regs}$ cluster frequencies were analyzed using binomial generalized linear mixed effects models. Log$_2$ fold-changes of comparisons with FDR < 0.05 are shown. Right: Average cluster frequencies across samples and annotation of the T$_{regs}$ clusters. **E** Correlation between the frequencies of pDC and ICOS⁺ T$_{regs}$ between 0 and 4 weeks post-infection. The frequencies were within-subject mean-centered and scaled to unit variance prior to pearson correlation analysis. Each point represents a sample from a particular time point, indicated using the colors. Lines and shaded band represent estimated mean and 95% confidence interval, respectively. **F** Neighborhood-level differential abundance analysis between 0 and 4 WPI using Milo. Left: Differentially abundant neighborhood of cells (SpatialFDR < 0.1) from Milo analysis comparing enrichment of neighborhoods of cells between 0 and 4 WPI. Red and blue points represent neighborhoods of cells and the layout of these points is determined by the position of the neighborhood index

cell in the FItSNE embedding from (**B**), which is depicted here by the gray points on the background. Color indicates log$_2$ fold change in enrichment, with red and blue color indicates enrichment at 4 and 0 WPI, respectively. Right: Close-up view of differentially abundant neighborhoods from the left panel, showing CRTH2⁺ neighborhoods (top right) of cells that are annotated as T$_H$2-like T$_{regs}$ (bottom right). **G** PHATE embedding of T$_H$2 cells (cluster 16) and T$_H$2-like T$_{regs}$ (T$_{regs}$ cluster 7) and overlays of marker expressions. **H, I** Frequencies of resting (CD45RA⁺) T$_{regs}$ (**H**) and CCR7⁻CTLA4⁺CD38^hiICOS⁺ effector T$_{regs}$ (T$_{regs}$ cluster 4) (**I**) in CHHIL (n = 4) and hookworm infected Flores subjects (n = 8). Lines connect paired samples. Vertical dashed lines demarcate the CHHIL and Flores data. Changes in frequencies relative to baseline (CHHIL subjects) or pre-deworming (Flores subjects) were tested using binomial linear mixed models with Dunnett's contrast. ***P < 0.001 for comparisons of post-infection CHHIL timepoints (4 to 104 WPI) against baseline (0 WPI). **J** Frequencies of T$_{regs}$ expressing TNFR2 (left) or TGF-β1 latency-associated peptide (LAP) (right) in CHHIL (n = 4) and hookworm-infected (HW⁺) Flores subjects (n = 3). Lines connect paired samples. Vertical dashed lines demarcate CHHIL and Flores donors. For the analysis of LAP expression, PBMC were stimulated using PMA/Ionomycin and then frequencies were background subtracted. For the analysis of TNFR2 expression, frequencies after in vitro culture in medium control (RPMI supplemented with 10% FCS) were analyzed. **K** Percentage (%) suppression of responder T cells (T$_{resp}$) proliferation by T$_{regs}$ in CHHIL and HW⁺ Flores residents (% suppression = (1 - (T$_{resp}$ proliferation with T$_{regs}$/T$_{regs}$ proliferation without T$_{regs}$)) x 100%). Left: Line plot showing % suppression across varying T$_{resp}$: T$_{regs}$ ratio as shown on the x-axis, summarised across individuals (CHHIL, n = 4; HW⁺ Flores, n = 3) and technical replicates (n = 3 per sample) within each group. Error bars indicate mean ± standard deviation. Right: Boxplots of percentage suppression by T$_{regs}$ in 1:1 T$_{resp}$:T$_{regs}$ ratio. Boxplots indicate the median and 25th/75th percentiles with whiskers extending to ±1.5 × IQR. Each data point represents the mean of 3 technical replicates. Differences in cell frequencies and percent suppression were tested using linear mixed-effects models with custom contrast comparing HW⁺ Flores against all CHHIL samples at 0, 12 and 104 WPI. T$_{regs}$ regulatory T cells, WPI week(s) post-infection, PC principal component, MSI median signal intensity, FI-tSNE FFT-accelerated Interpolation-based t-SNE. Source data are provided as a Source Data file.

---

CD38^hi cells from the CD4⁺T clusters analyzed in Fig. 4A. At 4 WPI, we observed a significant increase in CD38^hiCD4⁺T cell frequencies that remained elevated until 12 WPI and then normalized (Fig. 4C). Further exploration of these CD38^hiCD4⁺T cells revealed an increase in CD45RO⁺ cells that partially co-expressed ICOS, PD-1, or CTLA-4 (Fig. 4D), which could indicate the initial activation and expansion of antigen-specific cells, followed by a subsequent contraction in long-term infection.

Relating the emergence of antigen-specific cytokine responses to the occurrence of hookworm-related symptoms, it was observed that the peak of T$_H$1 and T$_H$2 cytokines corresponded to the time points when the majority of gastrointestinal symptoms were reported at 8 and 12 WPI, respectively (Table 1), suggesting that the occurrence of adverse events might be related to the increase of antigen-specific CD4⁺T cell cytokine responses.

Altogether, the controlled infection enabled us to map the dynamic changes of CD4⁺T cells in detail. We found that after hookworm infection, substantial T cell trafficking within the first 4 WPI and that an increase in CD4⁺T cells that are expressing CD38, a molecule strongly associated with in vivo T cell activation, precedes the development of hookworm antigen-specific T$_H$1 and T$_H$2 responses, which peak at 12 WPI.

## Discussion

Helminths, including hookworms, are experts at modulating the immune response of their hosts[19,20]. Through the combination of a controlled human hookworm infection model and high-dimensional immunophenotyping, using mass cytometry, we show that a first-in-life hookworm infection with 50 Na-L3 larvae lasting two years remodels the immune system characterized by alterations in circulating immune cells that comprised of pDC, basophils, T$_{regs}$, as well as

CD4⁺T cells (Fig. 5). These profiles were compared with data generated on samples collected from naturally infected individuals from Flores island in Indonesia, an area endemic for hookworm.

Notably, we found evidence of pDC modulation by hookworm, seen as increased pDC frequencies and reduced cytokine production after TLR-7/8 activation in both CHHIL participants and hookworm-infected subjects from Flores. pDC are activated by viral[21] or host-derived nucleic acids resulting from tissue injury[22], which might occur during the migration of hookworm larvae through their human host. When activated, pDC produce large amounts of type I interferon and express inducible co-stimulatory ligand (ICOS-L), among others. ICOS-L expression on pDC have been shown to induce T$_{regs}$ via ICOS binding[13,23–25]. In agreement, we found that the frequencies of pDC and ICOS-expressing T$_{regs}$ cluster were both increased between 0 and 4 WPI and their frequencies were highly correlated. Given the short half-life of pDC[26], the reduced cytokine production after in vitro TLR-7/8 stimulation might indicate perpetual modulation of pDC by the ongoing hookworm infection.

Another hallmark of T$_{regs}$ rearrangement among the CHHIL participants was the increase of resting T$_{regs}$ upon infection, which was also observed in hookworm-infected individuals from Flores. Resting T$_{regs}$ are long-lived and required for the maintenance of T$_{regs}$ populations[27], despite being less functional than effector T$_{regs}$[16,28]. Effector T$_{regs}$, on the other hand, may proliferate rapidly and migrate to tissues upon activation[29]. Therefore, given the relatively stable frequency of total T$_{regs}$ throughout infection, it appears that upon hookworm infection, resting T$_{regs}$ increase to maintain the circulating T$_{regs}$ pool at a relatively constant level while effector T$_{regs}$ migrate to peripheral organs. This is supported by a previous study by Croese and colleagues, who described increased T$_{regs}$ frequencies in the duodenum despite unaltered T$_{regs}$ frequencies in the peripheral blood of

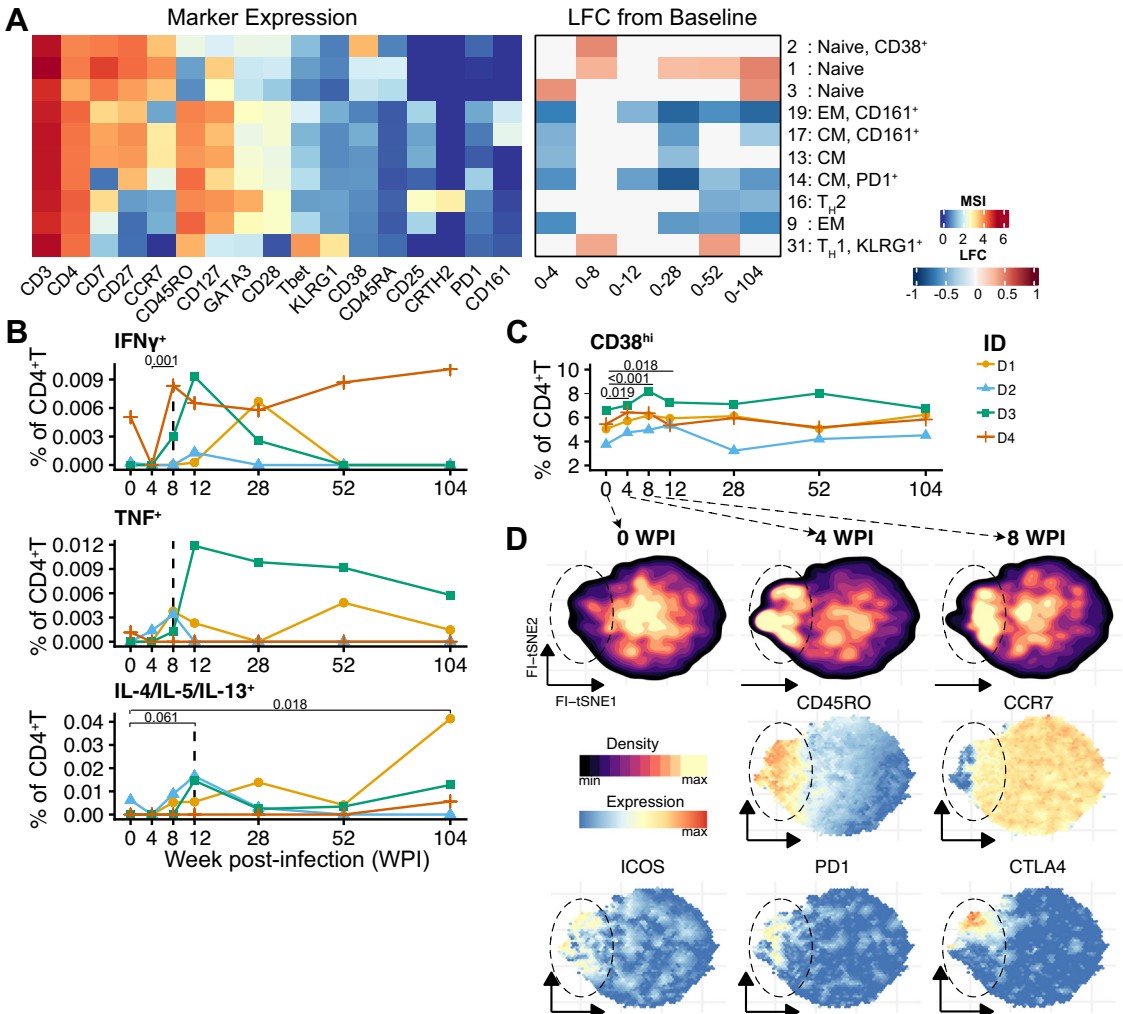

**Fig. 4 | Development of hookworm antigen-specific cytokine responses by CD4+T cells and the relevance of ex vivo CD38 expression on CD4+T cells.** **A** Heatmaps summarizing the changes in the frequencies CD4+T cell cluster frequencies, calculated relative to total CD45+ cells. Statistically significant (FDR < 0.05) log₂ fold change (LFC) of cell cluster frequencies relative to the baseline (left) or between consecutive time points (center). Heatmap of median signal intensity of selected markers (right). **B** Frequencies of CD4+T cells producing IFNγ, TNF or $T_H$2 (IL-4/IL-5/IL-13) cytokines upon stimulation with crude larval antigen of *Nippostrongylus brasiliensis*. Frequencies of cytokine-producing cells are background subtracted and negative values were set to zero, only for visualization. Lines connect paired samples. The dashed vertical lines highlight the time point at which the frequencies of cytokine-producing cells became detectable. Changes in frequencies were tested using linear mixed-effect models without adjustment for multiple comparisons. **C** Frequencies of manually gated CD38$^{hi}$CD4+T cells relative to total CD4+T cells from the ex vivo mass cytometry dataset. Lines connect paired samples. The changes in frequencies were tested using binomial generalized linear mixed effects models with Dunnett's contrasts. **D** FItSNE projection of CD38$^{hi}$CD4+T cells from (**C**) showing the embedding density (top) at 0, 4, and 8 WPI and the expression of selected markers (middle and bottom). TH1 T helper 1, $T_H$2 T helper 2, CM central memory, EM effector memory, WPI week(s) after infection. Source data are provided as a Source Data file.

celiac disease patients who had received experimental hookworm infection[30].

We also found an enrichment of $T_{regs}$ expressing CRTH2, a $T_H$2 cell marker, which according to our trajectory analysis could originate from activated conventional $T_H$2 cells transiently expressing Foxp3[15,31,32]. However, there is also the possibility that $T_{regs}$ start to express $T_H$2 cell markers. There is increasing interest in regulatory T cells expressing GATA3 or Tbet and their role in controlling adaptive immune responses under specific type 1 or type 2 inflammatory conditions in tissues[33–35]. Further studies are needed to establish the origin of these cells and their role in controlling type 2 inflammation during hookworm infections. For example, T-cell receptor (TCR) clonal tracing analysis of both $T_H$2 and $T_{regs}$ could show whether the TCR clonotype between these subsets overlaps and thereby provide evidence of developmental association[36].

Despite numerous changes in the $T_{regs}$ compartment after controlled infection, which in part resembled naturally occurring

infections in an endemic setting, we found differences in markers of $T_{regs}$ function: the suppressive capacity and frequencies of $T_{regs}$ expressing LAP and TNFR2 were higher in Flores residents than CHHIL participants. This could be explained by the higher and continuous exposure to helminths or the presence of co-infections common in an endemic area. This disparity in the $T_{regs}$ functional markers may help explain the modest efficacy, if any, of therapeutic experimental hookworm infections[37–40].

In agreement with previous studies[7], we found antigen-specific CD4+T cells expressing $T_H$1 (IFNγ and TNF) before $T_H$2 (IL-4, IL-5, IL-13) cytokines. We extended this to show that the frequencies of hookworm-specific $T_H$2 cells, but not $T_H$1 cells, remained elevated for up to two years after infection. Before the detection of hookworm-specific cytokine responses, we observed an increase in the frequencies of CD38$^{hi}$CD4+T cells. This is consistent with the observation that CD38 expression can accurately identify activated T cells in vivo[41]. In fact, within the CD38$^{hi}$CD4+T cells, we observed an enrichment of

**Table 1 | Reported adverse events and activity of antigen-specific cytokine-producing CD4⁺T cells per donor throughout the infection period**

| | 0 WPI | | | | 4 WPI | | | | 8 WPI | | | | 12 WPI | | | | 28 WPI | | | | 52 WPI | | | | 104 WPI | | | |
| --- | --- | --- | --- | --- | --- | --- | --- | --- | --- | --- | --- | --- | --- | --- | --- | --- | --- | --- | --- | --- | --- | --- | --- | --- | --- | --- | --- | --- |
| | 1 | 2 | 3 | 4 | 1 | 2 | 3 | 4 | 1 | 2 | 3 | 4 | 1 | 2 | 3 | 4 | 1 | 2 | 3 | 4 | 1 | 2 | 3 | 4 | 1 | 2 | 3 | 4 |
| **Adverse events[a]** | | | | | | | | | | | | | | | | | | | | | | | | | | | | |
| Rash | X | X | | | | | | | | | | | | | | | | | | | | | | | | | | |
| Itching | X | | X | X | | | X | X | | | | | | | | | | | | | | | | | | | | |
| Sore throat | | | | | | | | | | | | | | | | | | | | X | | | | | | | | |
| Cough | | | | | | | | | | | | | | | | | | | | X | | | | | | | | |
| Nausea | | | | | | | | | | | | | | X | | | | | | | | | | | | | | |
| Vomiting | | | | | | | | | | | | | X | | | | | | | | | | | | | | | |
| Bloating | | | | | | | | | | | X | X | | | | | | | | | | | | | | | | |
| Abdominal pain | | | | | | | | | X | X | X | | X | | | | | | | | | | | | | | | |
| Diarrhea | | | | | | | | | | | | | X | X | | | | | | | | | | | | | | |
| **Cytokines*** | | | | | | | | | | | | | | | | | | | | | | | | | | | | |
| IL-4/IL-5/IL-13 | | | | | | | | | + | + | + | + | + | + | + | | + | | | | | | | | | | | + |
| IFNγ | | | | | | | | | + | + | + | + | + | + | + | + | + | | | | | | + | | | | | + |
| TNF | | | | | | | + | + | + | + | + | + | + | + | + | | + | | | | | + | | | + | | | |

WPI week post-infection.

[a]Occurrence of adverse events at a particular time point (±1 week) for a particular donor is indicated with a cross mark (X). *Detection of antigen-specific CD4 + T cytokines is indicated with a plus symbol (+).

CD45RO⁺ cells expressing ICOS, PD1, and CTLA4, which implies strong activation of these CD4⁺T cells, parallel to the development of antigen-specific responses.

Although our study only involved four volunteers, we were able to detect significant patterns of change even after controlling for false discovery. However, several drawbacks need to be considered. The controlled hookworm infection was only given once, which is not representative of naturally occurring infections in endemic areas. Nonetheless, the comparison between CHHIL donors and Flores residents with chronic helminth infections revealed some similarities regarding the effects of hookworms on pDC and $T_{regs}$. Moreover, the focus on the circulating immune cells, which are more accessible than the duodenal epithelium, can result in the capture of signals that do not reflect the host's response to hookworm infection in tissues. Lastly, we did not include non-infected controls, which may help to disentangle fluctuations in the immune profile due to changes occurring over time. However, Liston and colleagues have shown that the immune phenotype within an individual is highly stable over time and may return to the initial equilibrium following perturbations, such as an acute infection or vaccination[42].

Taken together, we demonstrate that a single dose of first-in-life hookworm infection lasting two years resulted in a considerable remodeling of both innate and adaptive immune compartments. Comparison of immune cells from controlled infection donors with those from individuals living in the helminth-endemic area showed some resemblance between controlled and natural infections occurring in endemic areas, but also clear differences that can stem from continuous rather than bolus exposure to hookworm infections or other coinfections. Our study provides an important step in our understanding of the modulation of the human immune system by hookworms.

## Methods

### Controlled human hookworm infection trial

This trial was approved by the local institutional review board (protocol P17.001) and is registered at clinicalTrials.gov (NCT03126552). Healthy volunteers aged 18 to 45 years were recruited and provided written informed consent for enrolment in this trial[9]. Information regarding exclusion criteria, production of infective *N. americanus* L3 larvae, and procedure of infection have been extensively described elsewhere. Briefly, the exclusion criteria included, but were not limited to, body mass index below 18 or above 30, iron deficiency anemia, positive fecal tests for soil-transmitted helminths, albendazole contraindications, or planned travel to a hookworm-endemic area.

*Necator americanus* L3 larvae were produced according to the principles of good manufacturing practice. The fecal specimen containing *N. americanus* eggs was cultured for seven days before larvae harvest. The identity and viability of the infectious larvae were confirmed by PCR and microscopy. The larvae were used for infection within ten days after harvesting. Fifty *N. americanus* L3 larvae were dispensed within 15 min after preparation onto four gauzes, which were applied to the dorsal side of volunteers' upper arms and calves, respectively, for 60 min. Adverse events were recorded and the total eosinophil count and hemoglobin level were measured weekly during the first twelve weeks and six and twelve months after infection. For every adverse event, the time and date of onset and end, severity, and cause were recorded. The clinical trial physician classified adverse events into one of the following categories: unrelated or unlikely, possibly, or definitely related.

### Parasitological assays

Fecal samples were collected weekly from 5 WPI onward and checked for *N. americanus* eggs by Kato-Katz. Two microscopy slides per fecal sample were prepared and both were read by two different

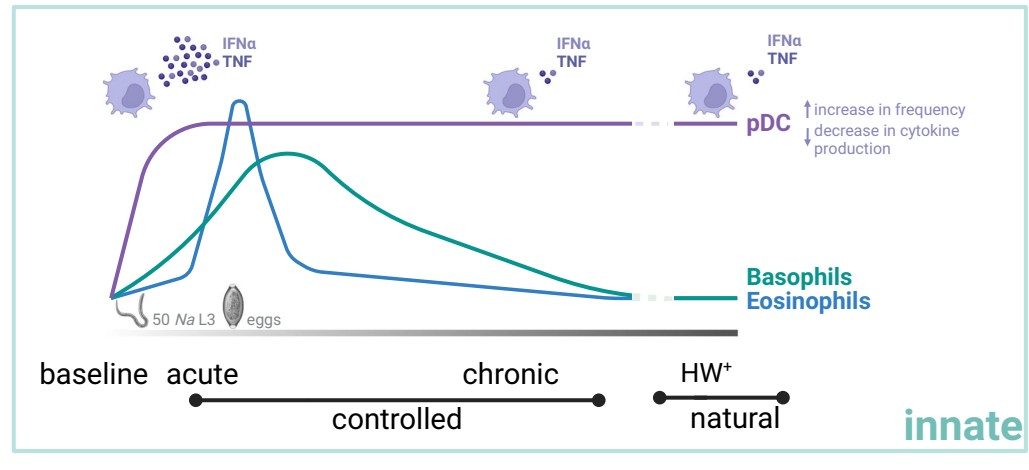

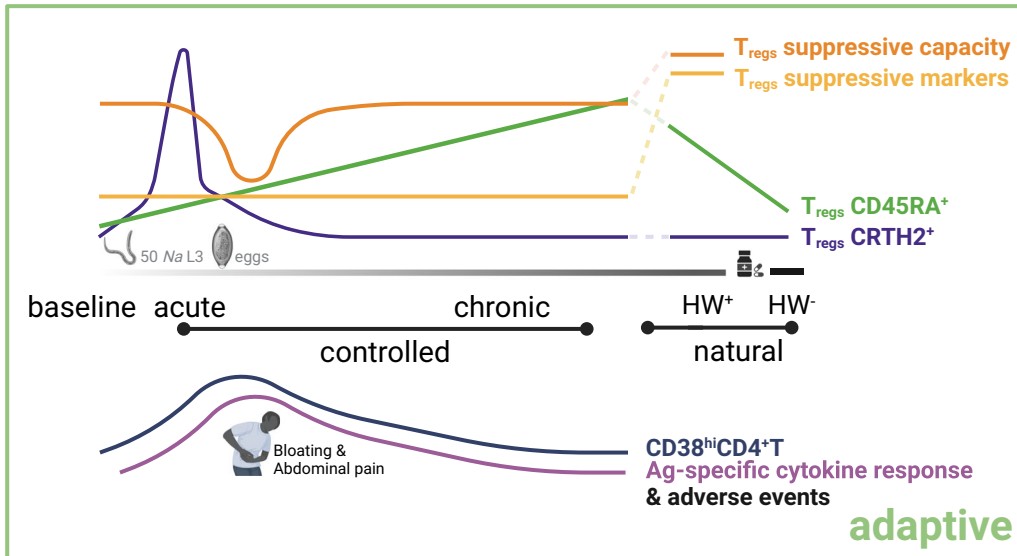

**Fig. 5 | Schematic summary of immune cell remodeling by human hookworm parasites in controlled and natural infection settings.** Top: trajectories of innate immune cells. Upon hookworm infection, pDC frequencies increased early and persisted through to chronicity, while their cytokine production in response to TLR-7/8 activation decreased steadily, resembling those observed in HW+ individuals in natural settings. The frequencies of basophils and eosinophils peaked around the time hookworm eggs could be detected and declined to a range comparable to that of HW+ individuals in natural settings. Bottom: Trajectories of adaptive immune cells. The frequency of CRTH2+ Tregs increased early after infection, however, prior to the detection of hookworm eggs, their frequency declined, while the frequencies of CD4+T cells expressing CD38 and those producing cytokines upon hookworm antigen stimulation increased between 8 and 12 WPI.

Notably, these series of events were accompanied by the occurrence of gastrointestinal adverse events, including bloating and abdominal pain. A dip in Tregs suppressive capacity was also observed during this same period. CD45RA+ Tregs frequencies steadily increased upon infection, eventually reaching a comparable range to those observed in HW+ individuals in natural infection settings, but declined upon anthelmintic treatment. Finally, Tregs suppressive markers and capacity were higher in HW+ individuals in natural infection settings. Lines represent chronological progression of immune parameters; line colors indicate distinct immune parameters. HW± hookworm-positive/-negative, pDC plasmacytoid dendritic cells, Tregs regulatory T cells, Ag antigen. This figure is created with BioRender.com under a Creative Commons Attribution-NonCommercial-NoDerivs 4.0 International license.

technicians. The total number of counted eggs was then multiplied by 20 and expressed as eggs per gram of feces (EPG).

## PBMC cryopreservation
Heparinized venous blood was diluted 1:1 with Hanks' balanced salt solution (HBSS; Invitrogen, 24020-133), containing penicillin G sodium (100 U/ml) and streptomycin (100 U/ml). PBMC were isolated using 1.077 Ficol density gradient centrifugation. PBMC were washed twice with HBSS and then cryopreserved in RPMI 1640 (Invitrogen, 42401-042) containing 20% heat-inactivated fetal calf serum (FCS; Bodinco, CP15-1439) and 10% dimethylsulfoxide (DMSO; Merck Millipore), 1 mM pyruvate, 2 mM L-glutamine, penicillin G, and streptomycin. Cryovials containing cell suspension were transferred to a Mr. Frosty freezing container (Thermo Fisher), which was placed

in a −80 °C freezer for four to 20 h before long-term storage in liquid nitrogen.

## Deworming of Flores residents chronically infected with helminths
The Flores individuals were part of the SugarSPIN trial, a household-based cluster-randomized double-blind deworming trial conducted in Nangapanda, Ende district of Flores Island, Indonesia[43]. The trial is registered as a clinical trial at the ISRCTN registry (reference no. ISRCTN75636394). Written informed consent was obtained from all participants before the study. Fifty-six helminth-infected individuals were recruited and treated with 400 mg of albendazole for three consecutive days. The anthelmintic regimen was administered every three months for a total of four rounds. Blood and stool samples were

collected before the start of the anthelmintic regimen and at the end of the four rounds.

Of the ten Flores individuals that were analyzed using mass cytometry in our previous study[10], eight individuals were included in this study based on the presence of hookworm infection (Table S4). Three additional Flores residents with hookworm infection were selected for intracellular cytokine staining assays (Table S4). Details on PBMC cryopreservation, as well as on mass cytometry antibody staining, data acquisition, and data analysis have been described elsewhere[10]. The mass cytometry panel used to analyze the Flores samples was similar to the panel used for the CHHIL study (Table S1) except for CD45RA ([113]CD-labelled, HI100 clone, eBioscience, Cat# 83-0458-42, 1:100 dilution) and that the Flores samples were not barcoded.

### Hookworm crude antigen preparation

*Nippostrongylus brasiliensis* (rodent hookworm) adult worms maintained in Wistar rats were kindly provided by Dr. Benjamin Dewals. Briefly, rats were infected subcutaneously with 5000 L3 larvae purified from fecal cultures. At day eight post-infection, the rats were euthanized, and the intestine recovered for isolation of adult worms in Hank's buffer using a modified Baermann's apparatus[44]. The worms were then decanted from excess media and stored at −80 °C. Adult worm crude antigen extract was prepared by lyophilization, homogenization, and sonication on ice. The antigen was then centrifuged and filter-sterilized. Antigen concentration was measured using a BCA assay kit (Pierce BCA Protein Assay Kit, Thermo Scientific) following the manufacturer's instructions. Lastly, to check for lipopolysaccharide (LPS) contamination, human embryonic kidney cells (HEK) were stimulated with worm antigen, and IL-8 was measured in the supernatant by ELISA. The antigen extract was then aliquoted and stored at −70 °C.

### Intracellular cytokine staining assay with flow cytometry
**Hookworm-specific responses**
**Cell culture.** Cells were thawed, washed in RPMI 1640 supplemented with 100 U/ml penicillin, 100 μg/ml streptomycin, 1 mM pyruvate, 2 mM glutamate, and 10% FCS, and adjusted to a concentration of $5 \times 10^6$ cells/ml. Co-stimuli αCD28 (BD Bioscience) and αCD49d (BD Bioscience) were added at a concentration of 1 μg/ml. Cells were then stimulated with 5 μg/ml antigen extract, 200 ng/ml staphylococcal enterotoxin B (SEB; Sigma-Aldrich), and 10% FCS/RPMI for 24 h at 37 °C under 5% $CO_2$. At the last four hours of stimulation, 10 μg/ml brefeldin A (Sigma-Aldrich) was added.

**Flow cytometry antibody staining.** After stimulation, cells were then washed twice in phosphate-buffered saline (PBS), stained for viable cells with LIVE/DEAD™ Fixable Aqua (Thermofisher), and fixed with 1.9% paraformaldehyde (Sigma-Aldrich) in PBS. Subsequently, cells were washed in FACS buffer (0.5% BSA in PBS, Roche, and 2 mM EDTA, Sigma-Aldrich) and then permeabilized with eBioscience™ Permeabilization Buffer (ThermoFisher). Cells were stained in a 96-well V-bottom plate with 50 μL of flow cytometry panel antibody mixture (Table S3) diluted in eBioscience™ permeabilization buffer (ThermoFisher) with 1% human Fc receptor blocker (eBioscience) at 4 °C for 30 min. Cells were then resuspended in the FACS buffer before acquisition.

**Cell acquisition.** Stained cells were acquired with a FACSCanto II flow cytometer (BD Biosciences). The compensation matrix was set using single-stained compensation beads (BD™ CompBead). FCS files were analyzed with the FlowJo v10 software (BD Life Sciences), where positions of gates were guided with fluorescence-minus-one (FMO) or unstimulated controls.

### Polyclonal responses to PMA/Ionomycin and R-848
**Cell culture.** PBMC were thawed as described above. For assessment of intracellular cytokine expression via spectral cytometry, cells were washed in RPMI 1640 (ThermoFisher), supplemented with 100 U/ml penicillin, 100 μg/ml streptomycin, 1 mM pyruvate, 2 mM glutamate, and 10% FCS (Greiner Bio-One), and adjusted to a concentration of $1 \times 10^6$ cells/ml. Cells were then resuspended in 100 μl RPMI + 10% FCS and stimulated for 6 h with PMA (100 ng/ml, Sigma-Aldrich) and ionomycin (1 μg/ml, Sigma-Aldrich) at 37 °C under 5% $CO_2$. After 4 h of stimulation, 10 μg/ml Brefeldin A (Sigma-Aldrich) was added.

**Flow cytometry antibody staining.** After stimulation, cells were washed twice in phosphate-buffered saline (PBS), stained for viable cells with LIVE/DEAD™ Fixable Blue (Thermofisher), washed again twice in FACS buffer (PBS supplemented with 0.5% BSA, Roche) and 2 mM EDTA (Sigma-Aldrich), and incubated with Human TruStain FcX™ (BioLegend) and True-Stain Monocyte Blocker™ (BioLegend) according to the manufacturer's instruction for 5 min at room temperature (RT). The antibody surface cocktail, prepared in Brilliant Stain Buffer Plus (BD Biosciences) was added to the cells and incubated for 30 min at RT. The list of antibodies can be found in Table S3. Cells were then washed twice in FACS buffer and afterward fixed and permeabilized with the eBioscience™ Foxp3 Transcription Factor Staining Buffer Set (ThermoFisher) for 30 min at 4 °C. Subsequently, cells were washed twice with the Permeabilization buffer from the eBioscience™ Foxp3 Transcription Factor Staining Buffer Set before being incubated with Human TruStain FcX™ and True-Stain Monocyte Blocker™ for 5 min at 4 °C and then stained with the intracellular/intranuclear antibody cocktail for 30 min at 4 °C (Table S3). Lastly, cells were washed with eBioscience™ Permeabilization buffer followed by another wash in FACS buffer. All centrifugation steps before fixation were performed at 300 g at room temperature (RT) and after fixation at 800 g at 4 °C.

**Cell acquisition.** For acquisition, cells were resuspended in FACS buffer and acquired on a 5L-Cytek Aurora instrument at the Leiden University Medical Center Flow Cytometry Core Facility (https://www.lumc.nl/research/facilities/fcf/) with the SpectroFlo® v2.2.0.3 software (Cytek Biosciences). As reference controls, an unstained cell sample and single-stain reference controls were used. Single-stain reference controls were either one million PBMC or 50 μl UltraComp eBeads™ (Invitrogen). All reference controls underwent the same protocol as the fully stained samples, including washes, buffers used, and fixation and permeabilization steps.

### Regulatory T cells (T_regs) suppression assay
CD4+CD25- responder T cells (T_resp) and CD4+CD25hi regulatory T cells (T_regs) were isolated via magnetic activated cell sorting (MACS) using the CD4+CD25+ Regulatory T Cell Isolation Kit (Miltenyi Biotec GmBH, 130-091-301). A two-step isolation was performed according to the manufacturer's recommendation, in which first CD4+ cells were isolated, followed by a separation step of CD4+CD25hi T_resp and CD4+CD25hi T_regs. After isolation, cells were rested overnight at 37 °C and 5% $CO_2$. CD4+CD25- T_resp ($5 \times 10^4$) was labeled with CFSE (Sigma-Aldrich, 21888-25MG-F) and co-cultured with an equal number (ratio of 1:1) or a decreasing number of CD4+CD25hi Tregs (ratios of 5:1 or 10:1). Co-cultures, T_resp alone (ratio of 1:0) or T_regs alone (ratio of 0:1) were stimulated with anti−CD2/CD3/CD28-coated beads at a bead to cell ratio of 1:1 (Miltenyi Biotec; 130-092-909). On day 5 of culture, cells were harvested, washed twice in phosphate-buffered saline (PBS), and stained for viability using the LIVEDEAD Blue Kit (Thermo Fisher Scientific, L23105). Subsequently, cells were washed twice in FACS buffer (0.5% BSA in PBS, Roche, and 2 mM EDTA, Sigma-Aldrich) and stained

with the surface antibody cocktail for 30 min at RT. The antibody surface cocktails were prepared in FACS buffer containing 10% Brilliant Stain Buffer Plus (BD Biosciences). The list of antibodies can be found in Table S3. Cells were then washed twice in FACS buffer, fixed, and permeabilized with the eBioscience™ FoxP3 Transcription Factor Staining Buffer Set (ThermoFisher) for 30 min at 4 °C. Subsequently, cells were washed twice with FACS buffer and stored overnight in the refrigerator. The next day, cells were washed with permeabilization buffer from the eBioscience™ FoxP3 Transcription Factor Staining Buffer Set before being incubated with the intracellular/intranuclear antibody cocktail for 30 min at 4 °C (Table S3). Lastly, cells were washed with eBioscience™ Permeabilization buffer followed by another wash in FACS buffer. All centrifugation steps before fixation were performed at 450 x g at RT and after fixation at 800 x g at 4 °C. The proliferation of viable responder T cells was determined by CFSE expression via flow cytometry in live singlet CD3$^+$CD4$^+$ T cells (Fig. S5 E). The percentage (%) of suppression of T$_{resp}$ proliferation by T$_{regs}$ was calculated as follows for each T$_{resp}$:T$_{regs}$ ratio: (1 − (T$_{resp}$ proliferation with T$_{regs}$ / T$_{resp}$ proliferation without T$_{regs}$)) x 100%.

### Ex vivo immunophenotyping with mass cytometry

Nine pre-conjugated metal-tagged antibodies were obtained from Fluidigm (Fluidigm, CA, USA) while 24 + 6 purified monoclonal antibodies were obtained from BioLegend, 2 from Miltenyi, and 1 from eBioscience and conjugated in-house using Maxpar® X8 Antibody Labeling Kits (Fluidigm) or MaxPar® MCP9 Antibody Labeling Kits (Fluidigm) according to the manufacturer's instructions. Based on titration performance, the in-house conjugated antibodies were stored in Candor PBS-based Antibody Stabilization solution (Candor Biosciences, Germany) with azide at 4 °C. The list of antibodies used is presented in Table S1.

**Sample preparation and mass cytometry antibody staining.** All centrifugations before fixation were performed at 400 g for 5 min at RT and after fixation at 800 g for 5 min at RT. PBMC were thawed as described above and $3.5 \times 10^6$ cells per sample were washed twice in 2 ml Maxpar® Cell Staining Buffer (CSB; Fluidigm). Next, cells were stained with 100 µl barcode antibodies in CBS for 30 min at RT. β−2 microglobulin barcodes (106 Cd, 110 Cd, 111 Cd, 112 Cd, and 114 Cd, 116 Cd; Table S1) were used in a 6-choose-2 combination to multiplex the samples. Samples were separated into two batches with 15 samples in each batch, including a replicate reference sample. Samples were then pooled accordingly, washed twice in CSB, and then 1 ml of rhodium-containing intercalator solution (Cell-ID™ Intercalator-Rh, Fluidigm, diluted 1:500 in CSB) was added for viability staining. To both batches 10 µl anti-Fcγ receptor (BioLegend) was added and incubated for 10 min, before cells were stained in a 200 µl surface antibody cocktail, containing the surface antibodies (Table S1) in CSB for 45 min at RT. Cells were then washed twice in CSB followed by permeabilization using Foxp3 transcription factor staining buffer (eBioscience). For this, cells were incubated for 45 min at 4 °C, protected from light, in a 2 ml working solution. After that, cells were washed twice in Foxp3 Permeabilization buffer (eBioscience) and stained in a 200 µl intracellular antibody cocktail, containing each of the intranuclear antibodies (Table S1), for 30 min at RT, protected from light. Subsequently, cells were washed once with 1x Permeabilization buffer and twice with CSB, before DNA staining with 1 ml of 1x iridium-containing intercalator solution (1000x Cell-ID™ Intercalator-Ir (Fluidigm), diluted 1:1000 in Maxpar® Fix and Perm Buffer) at 4 °C overnight. Lastly, cells were washed twice in 2 ml CSB, divided into samples containing $7.5 \times 10^6$ cells/ml in 1 ml 10% DMSO 20% FCS/RPMI, and prepared for storage in liquid nitrogen as described above.

**Mass cytometry cell acquisition.** On the day of acquisition, cells were thawed in 1 ml 20% FCS/RPMI and washed in 2 ml CSB. Data were

acquired using a Helios™ mass cytometer (Fluidigm). The instrument was tuned, calibrated, and cleaned before use according to the manufacturer's instructions. Briefly, argon gas was used to generate plasma and to nebulize the cell suspension. EQ beads were confirmed to have a median Eu153 intensity of over 1000 to ensure adequate mass sensitivity. Cell suspensions were acquired with a flow rate of 30 µl/minute. Immediately before the acquisition, cells were resuspended to a concentration of $0.75 \times 10^6$ cells/ml in Maxpar® Cell Acquisition Solution (Fluidigm) and supplemented with EQ™ Four Element Calibration Beads (Fluidigm) at a final dilution of 1:10. Cells were measured at an acquisition rate between 200 and 250 events/s. The resulting FCS files were normalized using the bead-based normalization tool of the Helios software version *6.5.358* (Fluidigm).

**Mass cytometry data analysis.** To obtain live single cells, we first performed automated gating of cells based on Gaussian-derived parameters (center, offset, residual, and width) using the cytofclean R package[45] followed by manual gating of live, single CD45$^+$ cells with DNA stain and event length in FlowJo™ V10 for Mac (BD Life Sciences). Live, singlet, immune cells were then debarcoded using the CATALYST R package[46]. The intensity of the B$_2$M-Cadmium barcode staining was well distinguished (Fig. S6A) and 91.6% of the cells were confidently debarcoded (Fig. S6B). Data were then arcsinh-transformed with a cofactor of 5. The analysis of the reference samples from both batches showed comparable marker intensities (Fig. S6C) and cluster abundances (Fig. S6D), all of which suggested negligible batch effects.

Unsupervised clustering of the whole dataset was performed using FlowSOM with a 20 × 20 SOM grid and 50 iterations (rlen) followed by consensus meta-clustering with ward.D2 linkage[47]. The resulting meta-clusters were further merged using hierarchical clustering and the optimal number of final clusters was assessed using gap statistic[48]. The clusters were annotated using expert knowledge of immune cell lineage and subset marker expressions.

For unsupervised clustering of a smaller subset of the data, we used Cytosplore+HSNE with default parameters[49]. The number of HSNE scales was determined by taking the log of the number of cells and rounding it up. Gaussian mean-shift clustering of the HSNE embedding was performed and then the resulting clusters were exported for further analysis in R.

Dimensionality reduction was performed using either FI-tSNE (FFT-accelerated Interpolation-based tSNE) or PHATE (Potential of Heat-diffusion for Affinity-based Trajectory Embedding). FI-tSNE scales tSNE for very large data sets[50] and uses optimized parameters and initialization that allow for a more accurate representation of the global structure of the data[51]. PHATE is a trajectory-preserving dimensionality reduction algorithm that is better suited for the analysis of cells in a continuum of expression states[17].

Neighborhood-level differential abundance level analysis of mass cytometry data was performed using the Milo algorithm[14]. Milo allows the identification of differentially abundant cells without having to discretize cells into clusters. To build the KNN graph, 15 principal components were used as input and the number of nearest neighbors (k) was tuned to achieve the recommended average neighborhood size (5 * number of samples). The sampling of the KNN graph was done by taking 10% of the graph vertices with a graph refinement scheme. Multiple testing correction of differentially abundant neighborhoods was done using graph-overlap weighting and the SpatialFDR cutoff was set at the 0.1 level.

Manual gating of mass cytometry data was performed using the OMIQ software (www.omiq.ai).

**Mass cytometry data integration.** Data integration was performed using the cyCombine R package[52]. Briefly, before the integration, the expression of every marker was converted to ranks, individually for

each data set. A SOM with a grid size of 6×6 was applied to the combined datasets. The SOM node labels were then assigned to the original expression values and ComBat batch correction was applied per cluster. The corrected values were capped to the input range for each marker.

## Statistical analysis

Statistical analysis was performed using the R statistical software version 4.0.2. Data wrangling and visualization were performed using the tidyverse R package[53]. Heatmaps were generated using the ComplexHeatmap R package[54]. A complete list of software and algorithms can be found in Table S5.

Multivariate analysis of changes in immune cell frequencies was performed using principal component analysis (PCA) and permutational multivariate analysis of variance (PERMANOVA). Multi-level PCA was performed using the mixOmics R package to account for repeated measures. PERMANOVA was performed using the adonis function of the vegan R package with 9999 permutations and Euclidean distance.

Comparisons of immune cell frequencies across time points were performed using binomial and gaussian linear mixed models for mass cytometry and intracellular cytokine staining data, respectively[55]. Post-hoc consecutive (comparing one timepoint against the previous one) and Dunnett's (comparing post-infection vs. baseline) contrasts were performed using the emmeans R package to obtain $P$-values and effect sizes.

Unless mentioned otherwise, $P$-values were adjusted for multiple testing using the Benjamini-Hochberg false discovery correction rate (FDR) method. To ensure a consistent p-value cutoff across comparisons, we applied the FDR correction on all comparisons simultaneously instead of within each comparison; this is similar to the 'global' $P$-value adjustment implemented in the limma R package[56]. Both FDR and alpha cutoffs were set at the 0.05 level.

## Reporting summary

Further information on research design is available in the Nature Portfolio Reporting Summary linked to this article.

## Data availability

Source data for figures are provided with this paper. Processed mass cytometry data is stored in Zenodo under restricted access (https://zenodo.org/records/10889855). Raw mass and flow cytometry data are available from corresponding authors upon request through a data transfer agreement. Restrictions on access are due to patient confidentiality issues; even pseudonymized data can be related back to individuals in some cases. Source data are provided with this paper.

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

## Acknowledgements

We would like to thank the Flow Cytometry Core Facility at Leiden University Medical Center for assistance with our flow and mass cytometry experiments. This study is part of the EDCTP2 programme supported by the European Union. The Controlled Human Hookworm Infection in Leiden (CHHIL) trial was funded by the Dioraphte Foundation and by NWO Spinoza prize of M.Y. M.D.M. is funded by the Indonesian Endowment Fund for Education (LPDP, Reference No. S-1598/LPDP.3/2016). B.G.D. is a Senior Research Associate of the Fonds de la Recherche Scientifique (F.R.S-FNRS).

## Author contributions

Conceptualization: M. Roestenberg, M. Yazdanbakhsh. Funding Acquisition: M. Roestenberg, M. Yazdanbakhsh. Clinical trial setup: M.A. Hoogerwerf, J.J. Janse, M.Roestenberg. Clinical trial samples collection: M.D. Manurung, Y. Kruize, J.J. Janse, T. Supali. Materials and reagents: B.G. Dewals, A. Loukas. Mass cytometry experiments: M.D. Manurung, F. Sonnet, M. König. Flow cytometry experiments: F. Sonnet, M. Coppola, L. de Bes. Data analysis and visualization: M.D. Manurung. Writing: M.D. Manurung, F. Sonnet, S.P. Jochems, M. Coppola, and M. Yazdanbakhsh. All authors critically reviewed and approved the final version for publication.

## Competing interests

The authors declare no competing interests.
