## [Peer Review File · Nature Communications]

Controlled human hookworm infection models plasmacytoid dendritic cells and regulatory T cells towards profiles seen in natural infections in endemic areasREVIEWER COMMENTS

Reviewer #1 (Remarks to the Author):

This is a major study into the changes of immune cell phenotypes in human helminth infection as measured in naïve volunteers exposed to the hookworm *Necator americanus*. This in itself is a considerable logistical feat and the authors have generated a unique dataset. The analysis goes far beyond any previous report, using high dimensionality immunophenotyping over a two-year period, and importantly compares the Leiden volunteers to residents of Flores, an Indonesian island with a high level of endemic hookworm infection.

The key findings are fascinating; first, there is an expansion in pDC (Fig 2B, 2D) which may be more tolerogenic and promote Tregs, and this is observed in both the volunteers and in Flores; remarkably, the pro-inflammatory potential (measured by Type I IFN and TNF) of the pDCs seems to be greatly diminished in long-term infections (Fig 2G).

The authors also find substantial changes within the Treg population (Fig 3C, D), which perhaps is most associated with susceptibility to helminth infection, and one change in particular (ICOS expression) is positively correlated with the pDC expansion noted above.

Finally, the authors also chart Th1/Th2 responses in their subjects, which are slower and less consistent in nature than the pDC/Treg arm of the response. Taken together, the manuscript presents a major body of new information with fascinating insights and leads into the immunology of human helminth infection

Specific Comments

1. The authors are able to subdivide the Treg population into 10 subsets; I am not convinced that these represent clear and discrete groups of cells, or if there is more of a continuum between them with cells switching expression of the defining markers. On the other hand, it seems to me that the GATA3+ “Tregs” are more likely to be Th2 as shown by the continuum in Figure 3G. The definition of “resting” and “effector” Tregs also seems uncertain, as once a new perturbation (in this case hookworm) is introduced.
2. The basophilia observed in the volunteers (Fig 2E) is extremely interesting; this peaks well before the Th2 cytokine response (Figure 4B); similarly eosinophilia (Fig 1 B) peaks early and declines sharply at the time Th2 cytokines are observed. What is driving the early Type 2 response? Is there an ILC2 response in early infection?
3. Did the authors perform any functional studies on the pDCs or the Tregs from volunteers after 104 weeks of infection? This would be useful validation of the assignments given to each phenotype.
4. To what extent does the T cell compartment show an exhausted or tolerant phenotype? The cluster assignments in Fig 4A show naïve, effector and memory subsets, but it might be expected in

chronic infection to find more anergised cells, and the decline in cytokine responses despite ongoing infection (Fig 4 B) is consistent with this.

Minor Points

- In Figure 1, the egg counts are only shown until Week 12, rather than to Week 104
- “Treg” is often used in the singular (eg “Treg is” on line 238) which reads awkwardly to this reviewer.
- Several of the analytical tools (cyCombine, Cytosplore, Milo etc) are not clearly explained in the text.

Reviewer #2 (Remarks to the Author):

The manuscript by Manurung et al describes a detailed phenotypic analysis of PBMC taken from 4 volunteers at multiple timepoints following experimental infection with hookworms over a 104 week period, comparisons to PBMC taken from individuals infected with hookworm from an endemic region have been made. Results from changes in the phenotypic composition of PBMC following a year of treatment for hookworm are also shown.

The manuscript is predominantly data from a panel of 37 antibodies used in a mass cytometry analysis of PBMC. Cytokine responses of pDCs to a TLR7/8 ligand stimulation and hookworm antigen specific T cell responses have also been determined.

The results support the author’s conclusions that following experimental infection there is a sustained alteration in a number of lineages in PBMC notably increases in pDC as well as Tregs which are further analysed and compared to PBMC from individuals infected in endemic areas. The authors conclude that the experimental infection of humans with a single dose of larvae induces phenotypic changes in lymphocyte sub populations that are reflected in PBMC from individuals in endemic areas there are some differences in T reg populations between the cohorts.

The results comprise a significant amount of careful flowcytometry with advanced analysis in order to interpret such a complex multi-dimension data set and trajectories with time post infection. It is also the case that the establishment of a baseline response in a controlled experimental infection and its comparison to natural infection (individuals in an endemic area which one has long-term infections) is important if the experimental model is to be developed into a trackable challenge model.

My concern is that the data is mostly descriptive, supports and confirms published observations by many other groups that have already determined the key effects on PBMC in those with hookworm infections. This is not to say that the results should not be publishable, but I question the level of novelty and general interest that would be expected for a Nature Communications paper. In

addition, as discussed by the author's, there are a number of caveats with regard to single dosing and a lack of normal health unchallenged controls in the experimental group.

The suggestion that Treg from individuals from endemic areas might be more functionally suppressive is interesting but based on only on phenotypic evidence of TNFR2 and LAP expression. For this level of publication, it might be expected to functional demonstrate differences in suppressive effector function between the experimental and endemic groups?

The analysis of hookworm specific T cell frequencies (Fig 4B) is not well described in the text eg one donor does not decrease IFN gamma responses with time, they are also the highest frequency response in addition the same donor does not make TNF, While donor 3 has the opposite phenotype with IFN gamma and TNF. Of concern is the frequency of these responses most of which are less than 0.01% of CD4 cells, this is a very low frequency indeed to measure by ICS analysis and may only represent just a few cells in the positive gate, were the number of media control cells analysed in each case the same as the number of cells analyse post antigen stimulation? It would also be reassuring if the data could be recapitulated with another technique eg ELISpot/FluroSpot. How does the frequency of antigen specific T cells in the experimental model compare to endemic infection? It is surprising that unlike in other figures this was not performed to provide another metric for the experimental model system.

Reviewer #3 (Remarks to the Author):

The study sets out to investigate the immunological changes following human hookworm infection. The approach utilized mass cytometry, in a manner similar to a previous study carried out by this group in naturally infected individuals (Flores Indonesia) which highlighted expansion of Th2 cells and T regulatory cells in the periphery. The study was meticulously and carefully carried out throughout. An important aspect that sets the current study apart from the previous study and most others in this area is the present study utilizes a controlled human infection with *Necator americanus* in Europeans as both a primary source of data and as a comparison to data from a naturally acquired infection (Flores Indonesia). Moreover, the study is followed over two-year period. Analysis is more extensive than the previous study and highlights some interesting observations and allows the authors to generate some new hypotheses. The new data adds granularity to the increased T reg frequency observed after hookworm infection and identifies an elevated number plasmacytoid dendritic cells and basophils and a number of changes in the controlled study were reflected, in part, in the naturally infected populations. The res-stimulation studies examining cytokine production by peripheral T cells following a combination of *Nippostrongylus* antigen and SEB added some information, but the data was quite variable and without naïve controls was difficult to assess. Also, it would have been useful to have more justification for the use of the parasite homogenate antigen in these studies. The very nature of the

study makes it difficult to be anything other than speculative in trying to understand possible underlying mechanisms of immunity. They are aware of the limits of this type of study. It will undoubtedly be informative to those researchers interested in intestinal helminth parasitism and emphasizes the feasibility and validity of such studies in this and other similar systems.

Point-by-Point Response to the Reviewers

Reviewer #1

This is a major study into the changes of immune cell phenotypes in human helminth infection as measured in naïve volunteers exposed to the hookworm *Necator americanus*. This in itself is a considerable logistical feat and the authors have generated a unique dataset. The analysis goes far beyond any previous report, using high dimensionality immunophenotyping over a two-year period, and importantly compares the Leiden volunteers to residents of Flores, an Indonesian island with a high level of endemic hookworm infection.

The key findings are fascinating; first, there is an expansion in pDC (Fig. 2B, 2D) which may be more tolerogenic and promote Tregs, and this is observed in both the volunteers and in Flores; remarkably, the pro-inflammatory potential (measured by Type I IFN and TNF) of the pDCs seems to be greatly diminished in long-term infections (Fig. 2G).

The authors also find substantial changes within the Treg population (Fig. 3C, D), which perhaps is most associated with susceptibility to helminth infection, and one change in particular (ICOS expression) is positively correlated with the pDC expansion noted above.

Finally, the authors also chart Th1/Th2 responses in their subjects, which are slower and less consistent in nature than the pDC/Treg arm of the response. Taken together, the manuscript presents a major body of new information with fascinating insights and leads into the immunology of human helminth infection

We thank the reviewer for their interest in our work and insightful comments.

Specific Comments

1. The authors are able to subdivide the Treg population into 10 subsets; I am not convinced that these represent clear and discrete groups of cells, or if there is more of a continuum between them with cells switching expression of the defining markers.

We thank the reviewer for raising this important consideration. Recent advances in high-dimensional single-cell analysis have revealed the coexistence of discrete and continuous variations between cell types and that a cell type can exist in different states (PMID 35868277), leading to the identification of an unprecedented number of clusters. For example, a previous mass cytometry study revealed 22 distinct subpopulations of T_{regs} (PMID 26223658).

Here, our main analysis aim was to unravel rare T_{regs} clusters associated with hookworm infection. To this end, we employed two complementary approaches: Cytosplore and miloR. Cytosplore facilitates an interactive exploration of mass cytometry data, enabling the identification of additional heterogeneity that might be

missed by other approaches (PMID 29170529). Using this method, we identified 10 Tregs clusters, six of which had statistically significant changes in abundance over the course of the controlled infection. Nonetheless, determining the number of clusters or cell types remains a difficult problem that ultimately depends on the researcher's decision; hence, our decision to complement our clustering-based differential analysis using the clustering-free miloR approach. Please note that the miloR package (PMID 34594043) allows differential abundance analysis without clustering, which was not possible prior to this algorithm. We believe that both approaches have their distinct benefits and trade-off, and their use could be viewed as complementary

On the other hand, it seems to me that the GATA3+ "Tregs" are more likely to be Th2 as shown by the continuum in Fig. 3G. The definition of "resting" and "effector" Tregs also seems uncertain, as once a new perturbation (in this case hookworm) is introduced.

We acknowledge the concerns of the reviewer. Therefore, we opted to use an alternative nomenclature for CD45RA⁺ T_{regs}, naïve T_{regs}, which was also used by the authors who discovered this subset (PMID 30587582).

2. The basophilia observed in the volunteers (Fig. 2E) is extremely interesting; this peaks well before the Th2 cytokine response (Fig. 4B); similarly eosinophilia (Fig. 1 B) peaks early and declines sharply at the time Th2 cytokines are observed. What is driving the early Type 2 response? Is there an ILC2 response in early infection?

We thank the reviewer for bringing this up and we believe there are many interesting associations that might help us understand how the immune system is shaped; however, we are focusing on the long-term impact of hookworm infection on the immune system and therefore we hope that the reviewer will agree that we cannot dwell on the many different insights that our study provides. However, we will publish the data in public data repositories, including Zenodo, for others to explore.

To address this question, we determined cell clusters associated with eosinophilia in the early time points (from baseline until 12 WPI). We performed a sparse partial least squares (sPLS) analysis (mixOmics R package, PMID 36308696) with mass cytometry cluster frequencies as the predictor and eosinophil counts as the response. Among the top-5 clusters in sPLS component one, we found that basophils were positively associated with eosinophilia (Fig. R1). In contrast, ILC2 were negatively associated with eosinophilia (Fig. R1). Indeed, differential abundance analysis showed that the ILC2 frequency significantly decreased at 8 WPI (Fig. S2 in the manuscript), coinciding with the peak of eosinophilia observed in these volunteers. This could possibly reflect their migration from the blood into the tissue where they could provide the trigger for eosinophilia.

Fig. R1. Top-5 loadings of sPLS component 1. sPLS was performed on cell clusters and eosinophil count data from the first 12 weeks after infection. Upon tuning the sPLS algorithm, two components were retained with 20 and 6 variables kept for the first and second component, respectively. sPLS was performed using the mixOmics R package. Bar colors indicate the direction of association between cluster frequencies and eosinophilia (blue: negative; red: positive).

- Did the authors perform any functional studies on the pDCs or the Tregs from volunteers after 104 weeks of infection? This would be useful validation of the assignments given to each phenotype.

Regarding pDC, we have shown that the frequencies of pDC producing IFN- α and TNF upon stimulation with R848, a TLR7-agonist (Fig. 2G in the revised manuscript). We showed that the frequencies of cytokine-producing pDCs, which were higher in CHHIL donors at baseline than in HW⁺ Flores donors, decreased over time upon infection, reaching frequencies comparable to HW⁺ Flores residents at 104 WPI.

Regarding T_{regs}, we have performed T_{regs} suppression assay to investigate T_{regs} capacity to suppress responder T cells proliferation, following the remarks from Reviewer #2 (Fig. R3). To this end, we reached out to a collaborator in Indonesia to access PBMCs from HW⁺ Flores residents so that we could compare their T_{regs} suppressive capacity against those observed in CHHIL donors. Briefly, we found that the current controlled hookworm infection trial setting did not result in a significant change in T_{regs} suppression capacity at 104 WPI compared to the baseline, and that T_{regs} of Flores residents with chronic hookworm infection had a higher suppressive capacity. A more elaborate explanation of our findings, including the experimental setup, has been provided in our response to Reviewer #2.

- To what extent does the T cell compartment show an exhausted or tolerant phenotype? The cluster assignments in Fig. 4A show naïve, effector and memory subsets, but it might be expected in chronic infection to find more anergised cells, and the decline in cytokine responses despite ongoing infection (Fig. 4 B) is consistent with this.

To this end, we analyzed the frequencies of CD4⁺T cells expressing exhaustion or senescence markers from the mass cytometry dataset and the frequencies of cytokine-producing CD4⁺T cells upon stimulation with Staphylococcal Enterotoxin B (SEB)—the latter was performed as the positive control for our antigen specific stimulation assay. We found that the frequency of CD4⁺T cells expressing CTLA4, PD1, or KLRG1 significantly decreased at 104 WPI upon controlled infection (Fig. R2 A). There were no statistically significant changes in the frequency of cytokine-producing CD4⁺T cells upon stimulation with SEB at 104 WPI (Fig. R2 B). Taken together, these data suggest that chronic hookworm infection in this setting did not result in increased expression of exhaustion markers or decreased cytokine response to a polyclonal stimulus.

Fig. R2.

(A) Changes in the frequency of cells expressing CTLA4, KLRG1, or PD1 in CD4⁺T cells. CD4⁺T cells from the mass cytometry dataset were manually gated for these markers. Frequencies were calculated relative to CD4⁺ T cells and are shown as log₂ fold-change from baseline.

(B) Change in the frequencies of cells expressing cytokines upon stimulation with Staphylococcal Enterotoxin B (SEB) in CD4⁺T cells relative to the baseline. P-values were obtained using linear mixed models. WPI, week post-infection.

Minor Points

- In Fig. 1, the egg counts are only shown until Week 12, rather than to Week 104

We have now substituted the figure accordingly to show egg counts up to 104 weeks post-infection (Fig. 1 B).

- “Treg” is often used in the singular (eg “Treg is” on line 238) which reads awkwardly to this reviewer.

We thank the reviewers for pointing this out, and we have adjusted the manuscript accordingly.

- Several of the analytical tools (cyCombine, Cytosplore, Milo etc) are not clearly explained in the text.

We have modified the following sections in the manuscript to better convey our intention in using these tools:

Line 134 “... as well as clustering using Cytosplore, which enables interactive exploration of mass cytometry datasets and identification of rare cell clusters (Fig. 3 B).”

Line 151 “To this end, we used Milo to perform differential abundance analysis on partially overlapping neighborhoods of cells¹⁴, enabling such analysis without having to discretize cells into computationally driven clusters.”

Reviewer #2

The manuscript by Manurung et al describes a detailed phenotypic analysis of PBMC taken from 4 volunteers at multiple timepoints following experimental infection with hookworms over a 104 week period, comparisons to PBMC taken from individuals infected with hookworm from an endemic region have been made. Results from changes in the phenotypic composition of PBMC following a year of treatment for hookworm are also shown.

The manuscript is predominantly data from a panel of 37 antibodies used in a mass cytometry analysis of PBMC. Cytokine responses of pDCs to a TLR7/8 ligand stimulation and hookworm antigen specific T cell responses have also been determined.

The results support the author’s conclusions that following experimental infection there is a sustained alteration in a number of lineages in PBMC notably increases in pDC as well as Tregs which are further analysed and compared to PBMC from individuals infected in endemic areas. The authors conclude that the experimental infection of humans with a single dose of larvae induces phenotypic changes in lymphocyte sub populations that are reflected in PBMC from individuals in endemic areas there are some differences in T reg populations between the cohorts.

The results comprise a significant amount of careful flowcytometry with advanced analysis in order to interpret such a complex multi-dimension data set and trajectories with time post-infection. It is also the case that the establishment of a baseline response in a controlled experimental infection and its comparison to natural infection (individuals in an endemic area which one has long-term infections) is important if the experimental model is to be developed into a trackable challenge model.

We thank the reviewer for the positive remarks and appreciation of our work.

My concern is that the data is mostly descriptive, supports and confirms published observations by many other groups that have already determined the key effects on PBMC in those with hookworm infections. This is not to say that the results should not be publishable, but I question the level of novelty and general interest that would be expected for a Nature Communications paper. In addition, as discussed by the author's, there are a number of caveats with regard to single dosing and a lack of normal health unchallenged controls in the experimental group.

The suggestion that Treg from individuals from endemic areas might be more functionally suppressive is interesting but based on only on phenotypic evidence of TNFR2 and LAP expression. For this level of publication, it might be expected to functional demonstrate differences in suppressive effector function between the experimental and endemic groups?

We thank the reviewer for this important remark. To this end, we performed an *in vitro* suppression assay with T_{regs} . We invested considerable time to ensure that we could work on samples from Indonesia from chronically infected individuals to perform the T_{regs} suppression assay. In addition, considerable time went into the development of the assay in our laboratory as we consider this an important point raised by the reviewer. We found that the T_{regs} of HW^+ Indonesians have a higher capacity to suppress responder T cell (T_{resp}) proliferation than CHHIL donors across various ranges of autologous $T_{resp}:T_{regs}$ co-culture ratios (Fig. R3 A). The difference was the strongest at 1:1 $T_{resp}:T_{regs}$ ratio in which we observed a significantly higher suppression of cell proliferation by Tregs among HW^+ Flores residents compared to CHHIL donors regardless of the timepoints ($P = 0.028$, Fig. R3 B).

Fig. R3. Percentage suppression of responder T cell proliferation by T_{regs} .

(A) Line plot showing percentage suppression (mean \pm SD) across varying $T_{resp}:T_{regs}$ ratios as shown on the x-axis, summarized across individuals and replicates within each group. $CD4^+CD25^-$ responder T cells (T_{resp}) and $CD4^+CD25^{hi}$ regulatory T cells (T_{regs}) were isolated via MACS using the $CD4^+CD25^+$ Regulatory T Cell Isolation Kit (Miltenyi Biotec GmbH, 130-091-301). A two-step isolation was performed according to the manufacturer's recommendation, in which first $CD4^+$ cells were isolated, followed by a separation step of $CD4^+CD25^{hi}$ T_{resp} and $CD4^+CD25^{hi}$ T_{regs} . After isolation, the cells were rested overnight at 37 °C and 5% CO_2 . $CD4^+CD25^-$ T_{resp} (5×10^4) was labeled with CFSE (Sigma-Aldrich, 21888-

25MG-F) and co-cultured with an equal number (ratio of 1:1) or a decreasing number of CD4⁺CD25^{hi} T_{regs} (ratios of 5:1 or 10:1). Co-cultures, T_{resp} alone (ratio of 1:0) or T_{regs} alone (ratio of 0:1) were stimulated with anti-CD2/CD3/CD28-coated beads at a bead to cell ratio of 1:1 (Miltenyi Biotec; 130-092-909). On day 5 of culture, cells were harvested, washed twice in phosphate-buffered saline (PBS), and stained for viability using the LIVEDEAD Blue Kit (Thermo Fisher Scientific, L23105). Subsequently, the cells were washed twice in FACS buffer (0.5% BSA in PBS, Roche, and 2 mM EDTA, Sigma-Aldrich) and stained with the surface antibody cocktail for 30 min at RT. The antibody surface cocktails were prepared in FACS buffer containing 10% Brilliant Stain Buffer Plus (BD Biosciences). The list of antibodies used can be found in Table S5. Cells were then washed twice in FACS buffer, fixed, and permeabilized with the eBioscience™ FoxP3 Transcription Factor Staining Buffer Set (ThermoFisher) for 30 min at 4 °C. Subsequently, the cells were washed twice with FACS buffer and stored overnight in the refrigerator. The next day, cells were washed with permeabilization buffer from the eBioscience™ FoxP3 Transcription Factor Staining Buffer Set before being incubated with the intracellular/intranuclear antibody cocktail for 30 min at 4 °C (Table S5). Finally, the cells were washed with eBioscience™ Permeabilization buffer followed by another wash in FACS buffer. All centrifugation steps before fixation were performed at 450x g at RT and after fixation at 800x g at 4 °C. The proliferation of viable responder T cells was determined by CFSE expression via flow cytometry in live, singlet CD3⁺CD4⁺T cells (Fig. S5 E). Percentage (%) suppression of T_{resp} proliferation by T_{regs} were calculated as follows for each T_{resp}:T_{regs} ratio: $(1 - (T_{resp} \text{ proliferation with } T_{regs} / T_{resp} \text{ proliferation without } T_{regs})) \times 100\%$.

(B) Boxplots of percentage suppression of T_{resp} proliferation by T_{regs} in a 1:1 T_{resp}:T_{regs} ratio. Each data point represents the mean of 2-3 technical replicates. P-values were obtained using linear mixed models comparing HW⁺ Flores residents against all CHHIL donors at 0, 12, and 104 WPI with custom contrasts [1, -1/3, -1/3, -1/3]. T_{resp}, responder T cells; T_{regs}, regulatory T cells.

We have included the following text in our revised manuscript with tracked changes (starting from line 182), and we also included Fig. R3 in our revised manuscript as Fig. 3K:

“To determine differences in T_{regs} functional capacity, we analyzed the surface expression of tumor necrosis factor receptor 2 (TNFR2) and TGF-β1 latency-associated peptide (LAP) on T_{regs} and also percentage suppression of responder T cell (T_{resp}) proliferation with T_{regs} suppression assay. The frequency of T_{regs} expressing TNFR2 and LAP was higher in HW⁺ Flores residents than that in CHHIL participants at 104 WPI (Fig. 3 J and Fig. S4 B). The percentage suppression by T_{regs} was higher in HW⁺ Flores residents than in CHHIL donors, particularly at 1:1 a T_{resp}:T_{regs} ratio (Fig. 3 K, P = 0.028; Fig. S5 E).

We have also modified the following line in the Discussion section to reflect our findings (line 268, exact additions are highlighted in yellow):

*“...we found differences in markers of Tregs function: **the suppressive capacity and frequencies of Tregs expressing LAP and...**”*

Details of the methods were added in line 427 of the revised manuscript and gating strategy of CFSE staining in CD3⁺CD4⁺ T cells are shown in Fig. S5.

The analysis of hookworm specific T cell frequencies (Fig. 4B) is not well described in the text eg one donor does not decrease IFN gamma responses with time, they are also the highest frequency response in addition the same donor does not make TNF, While donor 3 has the opposite phenotype with IFN gamma and TNF. Of concern is the frequency of these responses most of which are less than 0.01% of CD4 cells, this is a very low frequency indeed to measure by ICS analysis and may only represent just a few cells in the positive gate, were the number of media control cells analysed in each case the same as the number of cells analyse post antigen stimulation? It would also be reassuring if the data could be recapitulated with another technique eg ELISpot/FluroSpot. How does the frequency of antigen specific T cells in the experimental model compare to endemic infection? It is surprising that unlike in other figures this was not performed to provide another metric for the experimental model system.

We thank the reviewer for bringing this point up. Not only in the literature (PMID14659906), but also our past collaborators in companies such as J&J have performed comparative tests between ELISPOT and intracellular cytokine assays to show that they correlate. However, we understand that the reviewer would like to see this in the current work. We have analyzed the frequencies of peripheral immune cells producing IFN γ upon stimulation with crude hookworm antigen using ELISPOT (Fig. R3 A). Here, we showed that ELISPOT could indeed recapitulate our flow cytometry findings and their correlation with ICS data. We found a higher frequency of cells producing IFN γ (representative T_H1 cytokine) or IL-5 (representative T_H2 cytokine) at 12 WPI compared to the baseline (Fig. R3 B, top panel). In addition, the frequencies of IFN γ using ELISPOT and ICS were correlated, although they fell short of statistical significance due to the low numbers of individuals (Spearman's Rho = 0.69, P = 0.06, Fig. R3 C). These data suggest that the low frequency of cytokine-producing CD4⁺ T cells from our flow cytometry analysis was less likely to be an artifactual finding and that our interpretation should hold.

Fig. R4. ELISPOT analysis of cytokine-producing peripheral immune cells upon stimulation with crude hookworm antigen.

(A) Representative images taken from the IFN γ ELISPOT assay. After thawing, PBMCs were rested overnight at 37 °C and 5% CO₂. The IFN γ ELISPOT assay (ELISPOT Pro kit, Mabtech, 3420-2HPW-2) was performed according to the manufacturer's recommendations. A cell density of 2 × 10⁵ cells/well was used. Cells were either stimulated with hookworm antigen (*N. brasiliensis* crude antigen extract 5 ug/ml) or left unstimulated at 37 °C and 5% CO₂ for 24h. The next day, cells were discarded and after washing the plate five times with PBS, wells

were incubated with the detection antibody (MT20D9-biotin) for 2h at RT. After again washing the plate five times with PBS, streptavidin-ALP was added to the wells for 1h at RT. Wells were then again washed five times with PBS, and ready-to-use NBT/BCIP tablets were used for the color reaction. The color reaction was stopped by rinsing the plate with cold water. After drying, the plate was analyzed using an automated ELISPOT reader (Bio-Sys GmbH, Germany).

(B) Frequency of cytokine-producing cells upon stimulation after background subtraction for ELISPOT (top panel) and ICS (bottom panel). Negative values were set to zero only for visualization. P-values were obtained using linear mixed model analysis.

© Scatter plot showing correlation between IFN γ ELISPOT (in SFU unit) and ICS (in % of CD4+T cells). The regression line fit is shown.

Reviewer #3

The study sets out to investigate the immunological changes following human hookworm infection. The approach utilized mass cytometry, in a manner similar to a previous study carried out by this group in naturally infected individuals (Flores Indonesia) which highlighted expansion of Th2 cells and T regulatory cells in the periphery. The study was meticulously and carefully carried out throughout. An important aspect that sets the current study apart from the previous study and most others in this area is the present study utilizes a controlled human infection with *Necator americanus* in Europeans as both a primary source of data and as a comparison to data from a naturally acquired infection (Flores Indonesia). Moreover, the study is followed over two-year period. Analysis is more extensive than the previous study and highlights some interesting observations and allows the authors to generate some new hypotheses. The new data adds granularity to the increased T reg frequency observed after hookworm infection and identifies an elevated number plasmacytoid dendritic cells and basophils and a number of changes in the controlled study were reflected, in part, in the naturally infected populations.

We thank the reviewer for the generous comments and interest in our work.

The res-stimulation studies examining cytokine production by peripheral T cells following a combination of *Nippostrongylus* antigen and SEB added some information, but the data was quite variable and without naïve controls was difficult to assess.

The reviewer has raised an important point about variability in the data, which we believe has been addressed in our response to Reviewer #2 by performing the ELISPOT assay. Briefly, we found that ELISPOT showed a trend of increasing frequencies of cells producing IFN γ upon crude hookworm antigen stimulation at 12 WPI compared to baseline (Fig. R4 B, $P = 0.074$), and ELISPOT was correlated (Fig. R4 C, Spearman's $Rho = 0.69$, $P = 0.06$) with the flow cytometry intracellular cytokine staining data, but given the small number of individuals in this trial, the P-value falls short of statistical significance.

Regarding naïve controls, we should note that all CHHIL participants were hookworm-naïve at baseline; thus, each individual served as their own negative control.

Also, it would have been useful to have more justification for the use of the parasite homogenate antigen in these studies.

To our knowledge, crude hookworm antigens have been widely used in functional studies by various research groups (PMID 22087344, 17576364).

The very nature of the study makes it difficult to be anything other than speculative in trying to understand possible underlying mechanisms of immunity. They are aware of the limits of this type of study. It will undoubtedly be informative to those researchers interested in intestinal helminth parasitism and emphasizes the feasibility and validity of such studies in this and other similar systems.

REVIEWERS' COMMENTS

Reviewer #1 (Remarks to the Author):

The authors have responded in a comprehensive manner to all the points raised by myself and the other reviewers. I have just a few comments for them to consider.

1. Regarding the earlier question of whether Treg are "resting" or "effector", I find the use of the term "naive" instead, even more confusing; although this term is used in the Ha 2019 paper (30587582) I don't think it has been widely adopted, and in any case what does it mean? If the Treg are self-reactive they can't be naive; if they are specific for exogenous specificities, how have they arisen or been selected? So I would suggest that "resting" and "effector" or "activated" is a clearer terminology.

2. A point which I should have noted before is that the authors' "hookworm extract" is not from the human parasite *Necator americanus*, but from a rodent parasite *Nippostrongylus*. Although *Nippostrongylus* is not actually a hookworm, the use of this material is logistically reasonable, but the authors should clarify this in the text and legend to Figure 4.

3. Could the authors compose a summary schematic or a graphical abstract that captures their key findings in a visual form? This could highlight the common features, and the differences, found in the CHHI and Indonesian study groups.

Reviewer #2 (Remarks to the Author):

I Thank the authors who have addressed my comments with additional experiments and I am content with the modification they have made to the manuscript.

Reviewer #3 (Remarks to the Author):

I am happy with the authors responses to my comments.

RESPONSE TO REVIEWERS' COMMENTS

We appreciate the reviewers for the interests and feedbacks that made this publication possible. Please find below our line-by-line response in blue.

Reviewer #1 (Remarks to the Author):

The authors have responded in a comprehensive manner to all the points raised by myself and the other reviewers. I have just a few comments for them to consider.

1. Regarding the earlier question of whether Treg are "resting" or "effector", I find the use of the term "naive" instead, even more confusing; although this term is used in the Ha 2019 paper (30587582) I don't think it has been widely adopted, and in any case what does it mean? If the Treg are self-reactive they can't be naive; if they are specific for exogenous specificities, how have they arisen or been selected? So I would suggest that "resting" and "effector" or "activated" is a clearer terminology.

We have reverted the terminology to "resting" and "effector" Tregs as well as explicitly stating the that these subsets were defined as either CD45RA⁺ or CD45RA⁻ T_{regs}, respectively.

2. A point which I should have noted before is that the authors' "hookworm extract" is not from the human parasite *Necator americanus*, but from a rodent parasite *Nippostrongylus*. Although *Nippostrongylus* is not actually a hookworm, the use of this material is logistically reasonable, but the authors should clarify this in the text and legend to Figure 4.

We have added this information into the text (line 203) and figure caption (line 868), as follows:

(Line 203) "...At baseline and 4 WPI, the CD4⁺ T cells cytokine responses to crude larval antigen of the rodent hookworm *Nippostrongylus brasiliensis* were negligible (Fig. 4 B)."

3. Could the authors compose a summary schematic or a graphical abstract that captures their key findings in a visual form? This could highlight the common features, and the differences, found in the CHHI and Indonesian study groups.

We have now added Figure 5 (referred in the manuscript text, line 234) into the manuscript to summarise the findings of our paper.

Reviewer #2 (Remarks to the Author):

I Thanks the authors who have addressed my comments with additional experiments and I am content with the modification they have made to the manuscript.

Reviewer #3 (Remarks to the Author):

I am happy with the authors responses to my comments.